# BETTER, FASTER: HARNESSING SELF-IMPROVEMENT IN LARGE REASONING MODELS

## ABSTRACT

While large reasoning models (LRMs) trained with explicit reasoning trajectories have demonstrated impressive performance, obtaining high-quality trajectories is often costly and time-consuming. Hence, recent literature introduces a *self-improvement* paradigm that enables LRMs to improve themselves by self-generating reasoning trajectories as training data without external supervision. However, we find that this method often falls short in complex reasoning tasks and even leads to model collapse. Through a series of preliminary analyses, we reveal two shortcomings of self-improvement in LRMs: (1) *data imbalance*, where most training samples are simple, but the challenging yet crucial samples are scarce; (2) *overthinking*, where many undesired samples with redundant and repetitive reasoning steps are used for self-training. To this end, we propose `HSIR`, which effectively **H**arnesses **S**elf-**I**mprovement in large **R**easoning models via two simple-yet-effective approaches. Specifically, `HSIR` introduces a *verify-then-exit* sampling strategy to mitigate data imbalance by efficiently collecting more accurate solutions for difficult queries, and designs an *Intrinsic Diversity* score to quantify overthinking and filter out the undesired solutions. We apply `HSIR` to various post-training paradigms, among which we further propose `H-GRPO`, an enhanced GRPO algorithm that leverages the intrinsic diversity as an external reward to encourage concise and diverse reasoning via reinforcement learning. Extensive results show that `HSIR` not only effectively enhances the reasoning performance, *i.e.*, bringing up to **+10.9%** average performance gains, but also significantly improves the reasoning efficiency by reducing up to **42.4%** relative inference overhead.

## 1 INTRODUCTION

Recently, post-training the large language models (LLMs) with explicit long chain-of-thought (CoT) reasoning trajectories has garnered significant attention (Li et al., 2025; Plaat et al., 2024; Xu et al., 2025). Owing to the scaling inference compute of long-CoT reasoning, large reasoning models (LRMs) can unleash their reasoning capabilities (*e.g.*, backtracking and self-correction) and achieve better performance in various reasoning tasks, such as mathematical reasoning (Shao et al., 2024) and medical reasoning (Chen et al., 2024c). However, the performance of LRMs highly relies on high-quality intermediate reasoning trajectories (Yang et al., 2025c), which are usually costly and time-consuming to obtain (Peng et al., 2025). In response to this issue, the "*self-improvement*" paradigm has emerged, *i.e.*, models iteratively improve themselves by using the self-generated reasoning trajectories as training data, thereby reducing their dependence on external supervision.

In the context of LRMs, several self-improvement approaches have been proposed to achieve better reasoning performance, such as STaR (Zelikman et al., 2022) and ReST$^{EM}$ (Singh et al., 2023). For instance, ReST$^{EM}$ first prompts the model to generate multiple reasoning paths for each question, then filters out the incorrect solutions, and finally fine-tunes the model using its own correct outputs. These methods can boost LRMs' performance on conventional reasoning tasks without external supervision. However, in our preliminary experiments (Figure 1), we found that they often fall short in complex reasoning tasks, *e.g.*, medical question-answering that requires the integration of specialized knowledge with detailed patient histories and comorbidities (Huang et al., 2025b). More seriously, they might suffer from *model collapse*, where models' performance degrades due to iterative self-training on model-generated data (Bertrand et al., 2024; Gerstgrasser et al., 2024). Through a series of analyses (§2.2), we reveal that these methods have two major shortcomings:

❶ *data imbalance*, *i.e.*, most training samples are relatively simple, whereas challenging yet crucial samples are scarce; ❷ *overthinking*, *i.e.*, many undesired solutions with redundant and repetitive reasoning steps are used for self-training, hindering models' accurate and concise reasoning.

Several prior studies also recognize these short-comings and attempt to address them. Specifically, an intuitive way to mitigate data imbalance is to collect more correct solutions for difficult queries. To achieve it, Tong et al. (2024) propose to allocate more trials to difficult queries and Ding et al. (2025) leverage additional signals (*e.g.*, answers) to guide the reasoning of LRMs. Although effective, they mainly rely on increased inference overhead to obtain new solutions, while overlooking the potential value of previously failed solutions. On the other hand, to alleviate overthinking, the key lies in quantifying the redundancy and repetitiveness of reasoning steps. Most existing works involve designing length-oriented metrics and simply regard shorter correct solutions as superior (Team et al., 2025; Munkhbat et al., 2025). While achieving

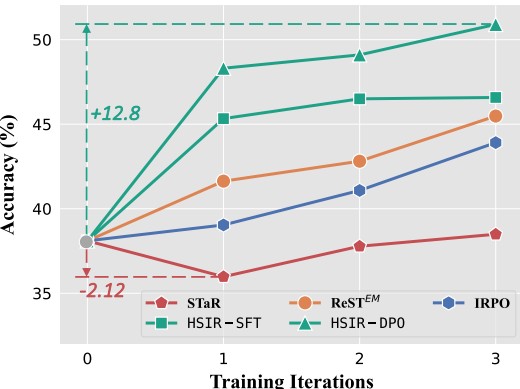

Figure 1: **Performance comparison** of Qwen2.5-1.5B models using different self-improvement post-training methods on the MedQA (Jin et al., 2021).

remarkable reasoning efficiency, overly emphasizing length reduction may hinder models' deep reasoning and lead to performance degradation (Dai et al., 2025). Thus, there arises a question: *can we explore a more effective self-improvement training method to make LRMs both better and faster?*

To achieve this goal, we propose **HSIR**, which effectively **H**arnesses **S**elf-**I**mprovement in large **R**easoning models via two simple-yet-effective approaches. First, to collect more correct responses for difficult queries, HSIR introduces a ***verify-then-exit*** (denoted as *VeriExit*) sampling strategy, which verifies the correctness of intermediate reasoning steps in the previously failed solution and self-truncates the reasoning once the current step arrives at the ground-truth answer. The motivation of *VeriExit* is that, within the failed solution, LRMs may have arrived at the correct answer during intermediate reasoning steps, yet ultimately failed to produce the accurate outcome due to reasoning deviation. Second, motivated by the intuition that a high similarity among intermediate reasoning steps often signals redundant or repetitive thinking, HSIR designs an ***Intrinsic Diversity*** score (denoted as *InDiv*) to quantify overthinking using the internal states of LRMs. In practice, *InDiv* performs an attention-aware eigenvalue analysis on the hidden representations of intermediate reasoning steps, where those with smaller eigenvalues are repetitive. Overall, by efficiently collecting more correct solutions and filtering out undesired overthinking solutions, HSIR can ensure the diversity and conciseness of training data, thus achieving better reasoning performance and efficiency.

We apply our HSIR to two iterative post-training paradigms: supervised fine-tuning (SFT) and preference learning. Extensive results on seven cutting-edge LLMs and two representative reasoning tasks, *i.e.*, medical reasoning and mathematical reasoning, show that our HSIR not only outperforms the other counterparts by a clear margin, but also effectively improves the reasoning efficiency. Furthermore, we expand our methods to the currently popular Reinforcement Learning from Verifiable Rewards (RLVR) training paradigm (Guo et al., 2025), and propose **H-GRPO**, an enhanced GRPO (Shao et al., 2024) algorithm that leverages the *InDiv* scores as an external reward to smoothly alleviate the overthinking. More comparative results prove the superiority of H-GRPO. Additionally, more in-depth analyses prove the effectiveness of HSIR's important components, and indicate that HSIR brings better model generalization. To summarize, our contributions are as follows:

- We reveal two major shortcomings of self-improvement in LRMs, and propose HSIR that leverages two simple-yet-effective approaches to alleviate them and make LRMs both better and faster.

- HSIR can be adopted to various post-training paradigms. Among which, we further expand it to the popular RLVR training paradigm, and propose H-GRPO that improves the GRPO by using our proposed *InDiv* scores as an external reward to encourage LRMs' diverse and concise reasoning.

- Extensive results show that HSIR can consistently and significantly improve the reasoning performance and efficiency for a diversity of LLMs, bringing up to **+10.9%** average performance gains and reducing up to **42.4%** relative inference overhead against the initial reasoning models.

## 2 RETHINKING SELF-IMPROVEMENT TRAINING IN LRMS

### 2.1 PRELIMINARIES

Considering that we have a base LLM $\mathcal{M}_{base}$, a small amount of seed data $\mathcal{S} = \{(x_i, r_i, y_i)\}_{i=1}^{N}$ and a larger unlabeled dataset $\mathcal{D} = \{(x_i, y_i)\}_{i=1}^{M} (M \gg N)$, where $x_i$ is the query, $r_i = [r_{i,1}, \ldots, r_{i,L}]$ is the corresponding reasoning trajectory with $L$ intermediate steps, and $y_i$ is the ground-truth answer. We first fine-tune $\mathcal{M}_{base}$ on $\mathcal{S}$ to make it have basic long-CoT reasoning ability, and denote the tuned model as $\mathcal{M}_0$. The goal of self-improvement is to enhance the reasoning performance of $\mathcal{M}_0$ by iteratively self-training using its own solutions on $\mathcal{D}$ over $T$ cycles. Specifically, let $\mathcal{M}_t$ denote the model at the $t$-th iteration ($t \in [1, T]$), the self-improvement training involves the following steps:

**Self-generation.** At $t$-th iteration, for each query $x_i \in \mathcal{D}$, we enforce the previous model $\mathcal{M}_{t-1}$ to generate multiple reasoning trajectories and their corresponding answers $\{(\hat{r}_i^k, \hat{y}_i^k)\}_{k=1}^{K}$, where $k \in [1, K]$ and $K$ denotes the total sampling times for each query. By doing so, we can obtain the self-generated dataset $\hat{\mathcal{D}}_t = \{(x_i, \hat{r}_i^k, \hat{y}_i^k) \,|\, x_i \in \mathcal{D}; k \in [1, K]\}$.

**Self-training.** The self-training process differs across various post-training paradigms. Specifically, during **SFT** training, the ground-truth answer $y_i$ is used to verify the correctness of candidate solutions $\{(r_i^k, \hat{y}_i^k)\}_{k=1}^{K}$, where only correct solutions with $\mathbb{I}(\hat{y}_i, y_i) = 1$ are filtered to form the pseudo-labeled dataset $\hat{\mathcal{D}}_t^{correct} = \{(x_i, \hat{r}_i^k, \hat{y}_i^k) \,|\, x_i \in \mathcal{D}; k \in [1, K]; \mathbb{I}(\hat{y}_i^k, y_i) = 1\}$. Notably, to alleviate the model collapse problem, we follow the prior studies (Alemohammad et al., 2024; Wang et al., 2024) and use the combination of the original clean seed dataset $\mathcal{S}$ and the pseudo-labeled dataset $\hat{\mathcal{D}}_t^{correct}$ as the final training dataset $\mathcal{D}_t = \mathcal{S} \cup \hat{\mathcal{D}}_t^{correct}$. Considering that continually fine-tuning $\mathcal{M}_{t-1}$ would lead to overfitting, we fine-tune the base model $\mathcal{M}_{base}$ on $\mathcal{D}_t$ to obtain the new model $\mathcal{M}_t$, following previous practice (Zelikman et al., 2022; Singh et al., 2023). In particular, we optimize $\mathcal{M}_t$ using the standard negative log likelihood (NLL) loss function:

$$\mathcal{L}_{\text{SFT}} = \mathbb{E}_{\mathcal{D}_t}\left[ -\log\frac{\mathcal{M}_\theta(\hat{r}_i^k, \hat{y}_i^k | x_i)}{|\hat{r}_i^k| + |\hat{y}_i^k|} \right], \tag{1}$$

where $\mathcal{M}_\theta$ initialized with $\mathcal{M}_{base}$ denotes the current tuned model that will become next model $\mathcal{M}_t$.

For the implementation of **preference learning**, we utilize a representative and effective algorithm, *i.e.*, Direct Preference Optimization (DPO) (Rafailov et al., 2023). Specifically, for each query $x_i \in \hat{\mathcal{D}}_t$, we split the candidate solutions into two sets: winner $\{(x_i, \hat{r}_i^{k_w}, \hat{y}_i^{k_w}) \,|\, \mathbb{I}(\hat{y}_i^{k_w}, y_i) = 1\}$ and loser $\{(x_i, \hat{r}_i^{k_l}, \hat{y}_i^{k_l}) \,|\, \mathbb{I}(\hat{y}_i^{k_l}, y_i) = 0\}$. Then, each winning solution and a randomly-selected losing solution are paired to construct the preference training set $\hat{\mathcal{D}}_t^{\text{pairs}} = \{(x_i, \hat{r}_i^{k_w}, \hat{y}_i^{k_w}), (x_i, \hat{r}_i^{k_l}, \hat{y}_i^{k_l}) \,|\, x_i \in \hat{\mathcal{D}}_t; k_w, k_l \in [1, K]\}$. Lastly, we can obtain the new model $\mathcal{M}_t$ by continually optimizing $\mathcal{M}_{t-1}$ on $\hat{\mathcal{D}}_t^{\text{pairs}}$. Inspired by Pang et al. (2024), we employ an enhanced DPO algorithm that combines the standard DPO loss function and NLL loss function on winning solutions to ensure the training stability, which is formulated as follows:

$$\mathcal{L}_{\text{DPO+NLL}} = \mathcal{L}_{\text{DPO}}(\hat{r}_i^{k_w}, \hat{y}_i^{k_w}, \hat{r}_i^{k_l}, \hat{y}_i^{k_l} | x_i) + \alpha_{nll} \cdot \mathcal{L}_{\text{NLL}}(\hat{r}_i^{k_w}, \hat{y}_i^{k_w} | x_i)$$

$$= \mathbb{E}_{\hat{\mathcal{D}}_t^{\text{pairs}}}\left[ -\log\sigma\left( f(\hat{r}_i^{k_w}, \hat{y}_i^{k_w} | x_i) - f(\hat{r}_i^{k_l}, \hat{y}_i^{k_l} | x_i) \right) - \alpha_{nll} \cdot \frac{\log\mathcal{M}_\theta(\hat{r}_i^{k_w}, \hat{y}_i^{k_w} | x_i)}{|\hat{r}_i^{k_w}| + |\hat{y}_i^{k_w}|} \right], \tag{2}$$

where $\sigma$ is the sigmoid function, $f(\cdot | x_i) = \beta\log\frac{\mathcal{M}_\theta(\cdot | x_i)}{\mathcal{M}_{t-1}(\cdot | x_i)}$, $\mathcal{M}_\theta$ is the policy model initialized with $\mathcal{M}_{t-1}$, $\alpha_{nll}$ and $\beta$ are coefficients that are empirically set to 0.5 and 0.1, respectively. Finally, we can obtain our next model $\mathcal{M}_t$, which will be used to generate data for the subsequent iteration.

### 2.2 EMPIRICAL ANALYSES

**Settings.** We conduct preliminary experiments by fine-tuning Qwen2.5-1.5B/3B/7B instruct models (Yang et al., 2024) on a challenging medical reasoning dataset, *i.e.*, MedQA (Jin et al., 2021). Specifically, since the original MedQA training set does not contain any reasoning trajectories, we prompt the proprietary DeepSeek-R1 to generate the seed reasoning data. By filtering out incorrect solutions, we ultimately obtained a new training set containing 9.3K reasoning samples. From this

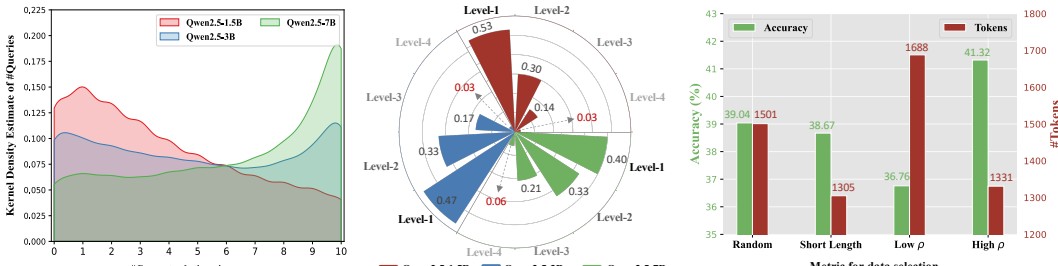

Figure 2: **Left**: Distribution of the number of correct solutions in a single query. **Middle**: Distribution of self-generated training samples with different difficulty levels, where level-1 means the simplest and level-4 means the most difficult. **Right**: Performance comparison between tuned Qwen2.5-1.5B models using different data selection methods. Here, all experiments are based on the MedQA task.

set, we randomly selected 1K as seed data $\mathcal{S}$, while treating the remaining samples as unlabeled data $\mathcal{D}$ (*i.e.*, without using their reasoning trajectories). For the implementation of self-improvement training, the number of iterations $T$ is set to 1, and the total sampling times $K$ is set to 10. Notably, $T = 1$ means that we perform the self-improvement training for one iteration, *i.e.*, from $\mathcal{M}_0$ to $\mathcal{M}_1$.

**Findings.** Through extensive analyses on the self-generated training samples, we found that there are two major problems: data imbalance and overthinking. Specifically,

❶ **Data Imbalance**: Figure 2 (**Left**) illustrates the distribution of the number of correct solutions in a query. As seen, there is a positive correlation between the number of correct solutions and model capabilities, where stronger models (*e.g.*, Qwen2.5-7B) can collect more accurate solutions for each query. However, for the difficult queries that are proven to be more crucial for further training (Liu et al., 2024), these models still struggle to collect sufficient correct solutions, thus leading to data imbalance. Specifically, although for the powerful Qwen2.5-7B, there are more than 500 queries that did not obtain any correct solutions. To have a close look, based on the number of correct solutions in a query, we evenly split the queries into four levels, where level-1 refers to the simplest queries obtaining the most correct solutions, and level-4 refers to the most difficult queries. Figure 2 (**Middle**) shows the distribution of self-generated training samples at different levels, indicating that **most training samples are relatively simple, whereas challenging yet crucial samples are scarce**.

❷ **Overthinking:** As a common issue in LRMs, overthinking usually leads to inefficient reasoning and suboptimal performance (Chen et al., 2024d). The key to alleviating overthinking lies in quantifying it via a fair and accurate metric. Motivated by the intuition that a high similarity among intermediate reasoning steps often indicates redundant or repetitive thinking, we introduce a **reasoning diversity metric** to measure the diversity of reasoning steps. As shown in Algorithm 1, for each reasoning trajectory $\hat{r}_i^k$, we first convert all intermediate reasoning steps $[\hat{r}_{i,1}^k, \ldots, \hat{r}_{i,L}^k]$ into sentence embeddings using the BGE-m3 model (Chen et al., 2024b), and then calculate the cosine distance between $\hat{r}_{i,l}^k$ and its nearest neighbor in the current subset. The reasoning steps with cosine distance below the threshold $\tau_{\text{sim}}$ are regarded as repetitive and are filtered out. The $\tau_{\text{sim}}$ is empirically set to 0.85 in our work. The metric $\rho$ is defined as the ratio of unfiltered steps to all steps:

$$\rho_i^k = \frac{|\mathcal{U}(\hat{r}_i^k)|}{|[\hat{r}_{i,1}^k, \ldots, \hat{r}_{i,L}^k]|}, \quad \rho_i^k \in (0, 1] \tag{3}$$

where $\mathcal{U}(\cdot)$ denotes the set of unfiltered steps. To verify the effectiveness of this metric, we conduct comparative experiments by using the solution with the highest and lowest $\rho_i^k$ for each query as self-training data, respectively. For reference, we also employ a random and a length-penalty method as baselines, *i.e.*, using a randomly selected solution and the shortest solution for self-training, respectively. Figure 2 (**Right**) shows the comparative results of Qwen2.5-1.5B models on the MedQA test set, using the average accuracy and number of output tokens as metrics. As seen, compared to using random solutions, self-training on solutions with low $\rho$ scores indeed results in more inference overhead and lower accuracy, while using the solutions with high $\rho$ scores can effectively alleviate this problem. Notably, although self-training on shorter solutions is also beneficial to improve reasoning efficiency, it would lead to performance degradation. These results suggest that **self-training with redundant and repetitive reasoning steps undermines the accuracy and conciseness of models' reasoning**, which is difficult to resolve effectively through length-penalty methods alone.

## 3 HARNESS SELF-IMPROVEMENT FOR BETTER AND FASTER REASONING

### 3.1 MOTIVATION AND INTUITION OF HSIR

To alleviate the above problems, we propose HSIR that harnesses self-improvement in LRMs via two simple-yet-effective approaches. First, to address data imbalance, we build upon the insight (Yang et al., 2025b; Dai et al., 2025) that failed solutions are not entirely incorrect but often contain valuable, partially correct reasoning steps before deviating. Instead of discarding these outputs, we introduce **VeriExit**, a novel **trajectory recycling** strategy. It efficiently salvages valid initial reasoning from failed attempts to generate correct solutions for difficult queries. This approach provides a significant efficiency gain over costly resampling from scratch. Second, to combat overthinking, our empirical analysis in §2.2 validates that reasoning diversity is a potent signal, yet the preliminary metric (Eq. 3) relies on costly external models. We propose a more elegant solution by harnessing the dense semantic information already present in a model's internal states. We introduce the **Intrinsic Diversity (InDiv) score**, an efficient metric that measures diversity directly from the models' hidden representations. This makes *InDiv* an **entirely intrinsic** measure that eliminates external dependencies and can be computed with minimal to zero overhead. Figure 3 illustrates the overview of our HSIR.

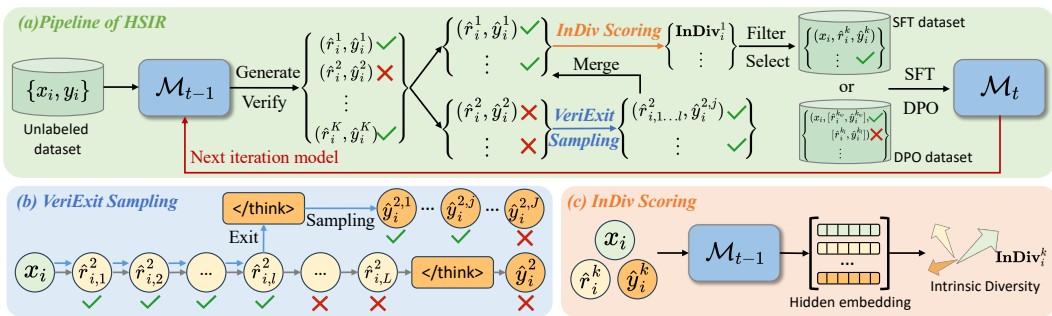

Figure 3: **(a)** Pipeline of self-improvement training with HSIR. After generating candidate solutions for each query, we first employ our **(b)** *VeriExit* sampling strategy to collect more accurate solutions for difficult queries, and then quantify the overthinking of correct solutions via our **(c)** *InDiv* metric. Lastly, the accurate, diverse, and concise solutions are selected for iterative SFT/DPO training.

### 3.2 IMPORTANT COMPONENTS OF HSIR

**Verify-then-Exit Decoding Strategy.** After obtaining the self-generated dataset $\hat{\mathcal{D}}_t$, we select the incorrect solutions from it to form a new set $\hat{\mathcal{D}}_t^{wrong} = \{(x_i, \hat{r}_i^k, \hat{y}_i^k) \,|\, x_i \in \mathcal{D}; \mathbb{I}(\hat{y}_i^k, y_i) = 0\}$. For each $\hat{r}_i^k \in \hat{\mathcal{D}}_t^{wrong}$, we verify the correctness of intermediate reasoning steps $[\hat{r}_{i,1}^k, \ldots, \hat{r}_{i,L}^k]$ by determining whether the $\hat{r}_{i,l}^k$ ($l \in [1, L]$) arrives at the ground-truth answer $y_i$[1], *e.g.*, explicitly mentioning "answer is $\{y_i\}$". Once $\hat{r}_{i,l}^k$ arrives at $y_i$, we truncate the subsequent reasoning steps and insert the exit prompt "\n\n</think>\n<answer>\n" at the truncated position. That is, we can obtain a new query "$x_i + [\hat{r}_{i,1}^k, \ldots, \hat{r}_{i,l}^k] + $ \n\n</think>\n<answer>\n", which is then fed into $\mathcal{M}_{t-1}$ to stop further reasoning and produce answers. Moreover, to ensure that $\mathcal{M}_{t-1}$ can output the correct answers, we sample $J$ times and collect the correct solutions $\hat{\mathcal{D}}_t^{VeriExit} = \{(x_i, \hat{r}_{i,1\ldots l}^k, \hat{y}_i^{k,j}) \,|\, \mathbb{I}(\hat{y}_i^{k,j}, y_i) = 1; k \in [1, K]; j \in [1, J]\}$. Notably, $J < K$, and such a sampling process will not lead to much inference overhead, compared to resampling the complete reasoning trajectories. Lastly, the $\hat{\mathcal{D}}_t^{VeriExit}$ is merged into the original correct solutions $\hat{\mathcal{D}}_t^{correct}$. If the total number of correct solutions for $x_i$ is larger than $K$, we randomly sample $K$ ones to maintain consistency between the training budgets of our method and vanilla self-improvement methods.

**Intrinsic Diversity Score.** To quantify overthinking, we leverage the LLMs' internal states to measure the semantic diversity of reasoning steps. Specifically, for each correct solution $(x_i, \hat{r}_i^k, \hat{y}_i^k)$, we obtain its hidden representation $\mathbf{H}_i^k \in \mathbb{R}^{d \times m}$ at the middle layer of $\mathcal{M}_{t-1}$, where $d$ is the dimension of hidden states and $m$ is the number of all tokens in the solution. We choose the middle

---

[1]There are several ways to achieve this, *e.g.*, text-matching, NLI-based and prompt-based methods. For simplicity and efficiency, we use the text-matching method by default. More analyses can be found in Appendix C.1.

layer as it encodes richer and more useful semantic information (Skean et al., 2025; Azaria & Mitchell, 2023; Liu et al., 2019). Inspired by the fact that eigenvalues of the covariance matrix can capture the divergence and correlation between different embeddings (Chen et al., 2024a), we calculate the eigenvalues of the cross-covariance for $\mathbf{H}_i^k$ to measure the diversity of intermediate reasoning steps:

$$\mathbf{\Sigma}_i^k = \mathbf{H}_i^{k\top} \cdot \mathbf{J}_d \cdot \mathbf{H}_i^k; \quad \mathbf{Eig}_i^k = \frac{1}{m} \log \det(\mathbf{\Sigma}_i^k) = \frac{1}{m} \sum_{u=1}^m \log(\lambda_{i,u}^k), \quad (4)$$

where $\mathbf{J}_d = \mathbf{I}_d - \frac{1}{d}\mathbf{1}_{d \times d}$ is the centering matrix, $\mathbf{I}_d \in \mathbb{R}^d$ is the identity matrix, $\mathbf{1}_{d \times d} \in \mathbb{R}^{d \times d}$ is the all-one matrix, $\det(\cdot)$ means the determinant of matrix, and $\{\lambda_{i,u}^k\}_{u=1}^m$ denotes the singular values of matrix $\mathbf{\Sigma}_i^k$. Furthermore, considering that some important tokens with higher attention weights might contribute more to the reasoning process, we enhance the above method via an attention-aware weighting mechanism. In practice, following the implementation of Su et al. (2024), let $Atten_{i,u}^k \in (0,1)$ denote the normalized maximum self-attention weight for $u$-th token ($u \in [1,m]$) among all self-attention heads, we can obtain our final reasoning diversity metric, denoted as **Intrinsic Diversity** score (*InDiv* in short) to distinguish it from Eq. 3, which is formulated as follows:

$$\mathbf{InDiv}_i^k = \sum_{u=1}^m \left[ Atten_{i,u}^k \cdot \log(\lambda_{i,u}^k) \right]; \quad \sum_{u=1}^m Atten_{i,u}^k = 1. \quad (5)$$

When the reasoning steps are repetitive and have similar semantics, the hidden representations will be highly correlated, and their semantic entropy and *InDiv* scores will be small (more analyses are in Appendix C.2). After calculating the *InDiv* scores of all candidate correct solutions $\{(x_i, \hat{r}_i^k, \hat{y}_i^k)\}_{k=1}^K$ for $x_i$, we filter the undesired ones with lower scores. Specifically, inspired by DeepSeek-r1 (Guo et al., 2025), we regularize the scores as $\overline{\mathbf{InDiv}}_i^k = \frac{\mathbf{InDiv}_i^k - \mathrm{mean}(\{\mathbf{InDiv}_i^1, \cdots, \mathbf{InDiv}_i^K\})}{\mathrm{std}(\{\mathbf{InDiv}_i^1, \cdots, \mathbf{InDiv}_i^K\})}$ and filter the solutions with scores below the threshold $\tau$. Finally, we can obtain more accurate, diverse, and concise reasoning data for effective self-training. The pseudo-code of HSIR is shown in Algorithm 2.

## 4 EXPERIMENTS

### 4.1 EXPERIMENTAL SETUP

**Tasks and Datasets.** We mainly assess the effectiveness of HSIR on both medical reasoning and mathematical reasoning tasks, using the MedQA (Jin et al., 2021) and GSM8K (Cobbe et al., 2021) datasets, respectively. For MedQA, we follow the settings in §2.2, and use 1K reasoning data distilled from DeepSeek-R1 as the seed data $\mathcal{S}$ and the other 8.3K data as the unlabeled dataset $\mathcal{D}$. Notably, the impact of different seed data can be found in Appendix C.4. For GSM8K, we use the 6.9K reasoning dataset released by CAMEL (Li et al., 2023). Similarly, 1K reasoning samples are randomly selected to form $\mathcal{S}$, while the remaining samples form $\mathcal{D}$. More specifically, the reasoning template is similar to that in DeepSeek-R1, *i.e.*, the reasoning process and answer are enclosed with <think></think> and <answer></answer> tags. Some training data examples are provided in Table 7 and 8. For evaluation, we report the zero-shot results on the original test sets of MedQA and GSM8K using the average accuracy and number of generated tokens as metrics. Since all models are evaluated on the same hardware, the number of generated tokens can reflect the wall-clock inference latency.

**Training Details.** We conduct main experiments using Qwen2.5-1.5B/3B/7B instruct models. To verify the generality of HSIR, we also evaluate it on another four instruct LLMs, including Qwen3-1.7B (Yang et al., 2025a), Phi-3.5-mini (Abdin et al., 2024), Mistral-7B (Jiang et al., 2023), and LLaMA3-8B (Dubey et al., 2024). During the implementation of HSIR, the sampling times $K$ and $J$ are set to 10 and 5, respectively. The sampling temperature is 1.0, and the maximum output length is 2,048. The filter threshold $\tau$ is set to -0.5. For the post-training of Qwen2.5 models, the self-improvement iteration $T$ is set to 3, but for the other LLMs, it is set to 1 due to limited computational resources. During inference, we use greedy decoding with a temperature of 0 for reproducibility. The maximum output length for all models is set to 4,096. More dataset and training details are shown in Appendix B, and the efficiency analysis of HSIR is shown in Appendix C.7.

**Baselines.** To verify the superiority of HSIR, we compare it with various training-based baselines:

- **SFT-Initial**: Standard fine-tuning $\mathcal{M}_{base}$ on the seed data $\mathcal{S}$ to obtain the initial SFT model $\mathcal{M}_0$.

Table 1: **Performance comparison between Qwen2.5 family models** using different training methods on MedQA and GSM8K. "|Train|" denotes the average number of training samples among all models and tasks, while "Overall" denotes the average accuracy and number of output tokens.

| Methods | \|Train\| Avg. | Qwen2.5-1.5B MedQA | GSM8K | Qwen2.5-3B MedQA | GSM8K | Qwen2.5-7B MedQA | GSM8K | Overall Accuracy | Tokens |
|---|---|---|---|---|---|---|---|---|---|
| SFT-Initial | 1.0K | 38.10 | 63.99 | 49.02 | 77.18 | 62.45 | 83.93 | 62.45 | 1,536 |
| SFT-Oracle | 8.1K | 46.58 | 71.57 | 58.68 | 84.31 | 73.99 | 87.79 | $70.49_{\uparrow 8.04}$ | $1,392_{\downarrow 9.4\%}$ |
| *(a) Iterative Self-improvement SFT Training* | | | | | | | | | |
| RFT | 132.2K | 42.42 | 71.19 | 54.60 | 83.55 | 64.89 | 87.87 | $67.42_{\uparrow 4.97}$ | $1292_{\downarrow 15.9\%}$ |
| ReGenesis | 127.2K | 44.46 | 66.26 | 50.67 | 79.91 | 62.22 | 89.99 | $65.59_{\uparrow 3.13}$ | 363 |
| STaR | | | | | | | | | |
|   Iteration 1 | 5.7K | 35.98 | 69.75 | 49.25 | 81.96 | 61.19 | 87.72 | $64.31_{\uparrow 1.86}$ | $1,379_{\downarrow 10.2\%}$ |
|   Iteration 2 | 6.1K | 37.78 | 70.96 | 50.82 | 81.27 | 61.04 | 88.48 | $65.06_{\uparrow 2.61}$ | $1,328_{\downarrow 13.5\%}$ |
|   Iteration 3 | 6.4K | 38.49 | 72.02 | 47.76 | 81.58 | 61.59 | 87.49 | $64.82_{\uparrow 2.37}$ | $1,288_{\downarrow 16.1\%}$ |
| ReST$^{EM}$ | | | | | | | | | |
|   Iteration 1 | 44.7K | 41.63 | 69.75 | 55.22 | 83.95 | 64.18 | 88.17 | $67.15_{\uparrow 4.70}$ | $1,268_{\downarrow 17.5\%}$ |
|   Iteration 2 | 51.0K | 42.81 | 74.45 | 56.25 | 85.13 | 65.28 | 90.22 | $69.02_{\uparrow 6.57}$ | $1,160_{\downarrow 24.5\%}$ |
|   Iteration 3 | 53.6K | 45.48 | 75.13 | 56.48 | 86.04 | 65.28 | 89.99 | $69.73_{\uparrow 7.28}$ | $1,114_{\downarrow 27.5\%}$ |
| `HSIR-SFT` (Ours) | | | | | | | | | |
|   Iteration 1 | 33.6K | 45.33 | 71.72 | 55.70 | 86.13 | 67.32 | 88.78 | $69.16_{\uparrow 6.71}$ | $1,075_{\downarrow 30.0\%}$ |
|   Iteration 2 | 36.6K | 46.50 | 76.04 | 56.32 | 86.51 | 67.87 | 90.83 | $70.68_{\uparrow 8.23}$ | $950_{\downarrow 38.1\%}$ |
|   Iteration 3 | 38.8K | **46.58** | **76.88** | **57.58** | **86.81** | **68.74** | **91.36** | $\mathbf{71.33}_{\uparrow 8.88}$ | $\mathbf{896}_{\downarrow 41.7\%}$ |
| *(b) Iterative Self-improvement DPO Training* | | | | | | | | | |
| IRPO | | | | | | | | | |
|   Iteration 1 | 22.7K | 39.04 | 70.51 | 47.99 | 85.37 | 64.26 | 91.43 | $66.43_{\uparrow 3.98}$ | $1,359_{\downarrow 11.5\%}$ |
|   Iteration 2 | 27.5K | 41.08 | 75.51 | 49.33 | 86.66 | 63.24 | 91.95 | $67.96_{\uparrow 5.51}$ | $1,294_{\downarrow 15.7\%}$ |
|   Iteration 3 | 20.5K | 43.91 | 75.36 | 49.10 | 87.19 | 60.57 | 91.87 | $68.00_{\uparrow 5.55}$ | $1,271_{\downarrow 17.2\%}$ |
| `HSIR-DPO` (Ours) | | | | | | | | | |
|   Iteration 1 | 17.9K | 48.31 | 75.36 | 55.77 | 87.04 | 67.32 | 91.36 | $70.86_{\uparrow 8.41}$ | $1,007_{\downarrow 34.4\%}$ |
|   Iteration 2 | 23.6K | 49.10 | 76.65 | 59.15 | 87.49 | 68.58 | 91.96 | $72.16_{\uparrow 9.70}$ | $921_{\downarrow 40.0\%}$ |
|   Iteration 3 | 19.6K | **50.90** | **78.09** | **60.64** | **87.53** | **70.46** | **92.49** | $\mathbf{73.35}_{\uparrow 10.90}$ | $\mathbf{885}_{\downarrow 42.4\%}$ |

- **SFT-Oracle**: Standard fine-tuning $\mathcal{M}_{base}$ on the combination of $\mathcal{S}$ and $\mathcal{D}$ with ground-truth reasoning trajectories, which can be considered as the upper bound of SFT training.

- **STaR** (Zelikman et al., 2022): Sampling a solution $(\hat{r}_i, \hat{y}_i)$ using greedy decoding for each query $x_i \in \mathcal{D}$, where the correct solutions are used to iteratively fine-tune the models.

- **ReST**$^{EM}$ (Singh et al., 2023): Extending STaR by sampling $K$ solutions $\{(\hat{r}_i^k, \hat{y}_i^k)\}_{k=1}^K$ for each query $x_i \in \mathcal{D}$, where all correct solutions are used for iterative self-improvement SFT training.

- **RFT** (Yuan et al., 2023): Similar to ReST$^{EM}$ but not iterative. To maintain consistent training budgets, we sample $T \times K$ candidate solutions $\{(\hat{r}_i^k, \hat{y}_i^k)\}_{k=1}^{T \times K}$ for each query $x_i \in \mathcal{D}$.

- **ReGenesis** (Peng et al., 2025): Prompting $\mathcal{M}_{base}$ to self-synthesize reasoning paths by converting general reasoning guidelines into task-specific ones, which are used for once self-training. Since it is not designed for long-CoT reasoning, we do not compare its reasoning efficiency.

- **IRPO** (Pang et al., 2024): Sampling $K$ solutions for each query, where both correct and incorrect solutions are paired to construct the preference data, allowing for iterative DPO training.

For all baselines, we keep a fixed data synthesis budget. Moreover, since our goal is to propose a self-improvement training method, we do not compare `HSIR` with inference-time methods in the main experiments. More comparisons with inference-time methods are shown in Appendix C.9.

## 4.2 MAIN RESULTS

**`HSIR` outperforms the other baseline methods across all post-training settings.** Table 1 reports the comparative results (%) of Qwen2.5 family models. As seen, self-improvement training on the relatively simple GSM8K task performs better against the challenging MedQA task, confirming that self-improvement methods fall short in complex reasoning tasks. More specifically, during SFT on MedQA, STaR struggles to enhance the LRMs' reasoning performance, and even leads to performance degradation, *e.g.*, from 38.10% to 35.98% in Qwen2.5-1.5B. By sampling more diverse solutions, RFT and ReST$^{EM}$ alleviate this problem, indicating the importance of self-training with diverse reasoning data. While in the DPO phase, the effectiveness of self-improvement is more

Table 2: **Performance comparison between the other models** using different self-improvement SFT methods on MedQA and GSM8K. Here, we perform the self-improvement training for one iteration.

| Methods | Qwen3-1.7B | | Phi-3.5-mini | | Mistral-7B | | LLaMA3-8B | | Overall | |
|---|---|---|---|---|---|---|---|---|---|---|
| | MedQA | GSM8K | MedQA | GSM8K | MedQA | GSM8K | MedQA | GSM8K | Accuracy | Tokens |
| SFT-Initial | 51.61 | 87.03 | 66.14 | 82.56 | 55.93 | 64.59 | 64.57 | 79.98 | 69.05 | 1,540 |
| SFT-Oracle | 55.77 | 87.72 | 74.07 | 87.95 | 70.15 | 79.08 | 73.76 | 86.73 | 76.90$_{\uparrow7.85}$ | 1,383$_{\downarrow10.2\%}$ |
| STaR | 52.87 | 86.96 | 65.99 | 86.28 | 53.57 | 68.54 | 59.63 | 80.59 | 69.30$_{\uparrow0.25}$ | 1,442$_{\downarrow6.3\%}$ |
| ReST$^{EM}$ | 54.67 | 88.61 | 67.64 | 87.86 | 58.13 | 72.71 | 67.79 | 81.20 | 72.33$_{\uparrow3.28}$ | 1,365$_{\downarrow11.4\%}$ |
| **HSIR-SFT** | **55.30** | **89.16** | **71.17** | **88.32** | **61.27** | **75.82** | **69.68** | **86.66** | **74.67**$_{\uparrow5.62}$ | **1,195**$_{\downarrow22.4\%}$ |

dependent on the quality of self-generated data, as preference learning is more data-sensitive. By selecting more diverse and concise self-generated data for training, our HSIR can effectively unleash the reasoning cabilities of LRMs, thus achieving better performance against the other baselines. For instance, our HSIR-DPO outperforms the vanilla iterative DPO method by a clear margin, *i.e.*, bringing +5.35% average performance gains among all Qwen2.5 models after three iterations.

**HSIR effectively improves the reasoning performance and efficiency of LRMs in both tasks.** In addition to the reasoning performance, we also evaluate the reasoning efficiency of LRMs by measuring the number of output tokens. From Table 1, it can be seen that nearly all self-improvement methods reduce the average output tokens. We conjecture that models can sometimes generate concise reasoning paths, which helps guide the efficient reasoning of models. Nevertheless, as shown in our preliminary analysis (§2.2), self-training with overthinking solutions would damage this effect and lead to suboptimal reasoning efficiency. Owing to our *InDiv* metric, we can filter these overthinking solutions and effectively improve reasoning efficiency by reducing up to 42.4% output tokens. These results confirm the significance of alleviating overthinking and prove the effectiveness of HSIR.

**HSIR brings consistent and significant performance gains among all model sizes and types.** Table 2 presents the results of other LRMs. Notably, due to limited computation resources, we only perform the SFT training using STaR, ReST$^{EM}$, and our HSIR for one iteration. As seen, HSIR continues to outperform the other baseline methods across all models. Specifically, in LLaMA3-8B, compared to powerful ReST$^{EM}$, HSIR achieves +1.89% and +5.46% performance gains for MedQA and GSM8K, respectively. Overall, HSIR brings +5.62% average performance gains and reduces 22.4% output tokens against the initial SFT models, showing its universality and superiority.

## 4.3 MORE ANALYSES

**Ablation Study.** In this part, we validate the important components of HSIR, *i.e.*, *VeriExit* sampling strategy and *InDiv* metric. Firstly, for the analysis of sampling strategy, we ignore the overthinking metric and do not perform the data filtering. To verify the effectiveness of *VeriExit*, we compare it with "-w Answer-driven" that uses the ground-truth answer to guide models' reasoning for collecting more correct solutions (Ding et al., 2025). Similarly, for the analysis of overthinking metric, we do not use extra sampling methods, and compare our *InDiv* with "-w Length-driven" that leverages the length of

Table 3: **Ablation study** on *VeriExit* and *InDiv*.

| Method | MedQA | | GSM8K | |
|---|---|---|---|---|
| | Accuracy | Tokens | Accuracy | Tokens |
| SFT-Initial | 38.10 | 1,779 | 63.99 | 1,666 |
| STaR | 35.98 | 1,651 | 69.75 | 1,377 |
| ReST$^{EM}$ | 41.63 | 1,424 | 69.75 | 1,382 |
| **HSIR-SFT (All)** | **45.33** | **1,064** | **71.72** | **1,181** |
| *(a) Analysis of sampling strategy (without data filtering)* | | | | |
| -w/ Answer-driven | 41.63 | 1,421 | 69.29 | 1,433 |
| **-w/ VeriExit (Ours)** | 44.46 | 1,293 | 70.74 | 1,361 |
| *(b) Analysis of overthinking metric (without extra sampling)* | | | | |
| -w/ Length-driven | 42.36 | 1,321 | 70.66 | 1,266 |
| **-w/ InDiv (Ours)** | 42.66 | 1,260 | 71.34 | 1,256 |

solutions as the metric. In practice, the candidate solutions with $\frac{\text{len}(\hat{r}_i^k) - \text{mean}(\{\text{len}(\hat{r}_i^1), \cdots, \text{len}(\hat{r}_i^K)\})}{\text{std}(\{\text{len}(\hat{r}_i^1), \cdots, \text{len}(\hat{r}_i^K)\})} > \tau_{\text{len}}$ are filtered, where $\text{len}(\hat{r}_i^k)$ denotes the length of $\hat{r}_i^k$ and $\tau_{\text{len}}$ is the length threshold set to 0.5 in this experiment. Table 3 reports the results of Qwen2.5-1.5B models after one iteration of self-improvement SFT training. Compared to the full HSIR, removing *VeriExit* or *InDiv* results in performance degradation, indicating their effectiveness. Moreover, our proposed methods consistently perform better than their counterparts. For instance, *VeriExit* outperforms the "-w Answer-driven" by 2.14% average performance gains. These comparative results demonstrate the superiority of *VeriExit* and *InDiv*.

**Expand to GRPO Training.** In addition to SFT and DPO training, reinforcement learning from verifiable rewards via the GRPO (Shao et al., 2024) algorithm is also a popular and effective way to

enhance LRMs' reasoning performance. Instead of explicitly supervising the reasoning trajectory, GRPO enables LRMs to learn from free exploration via outcome rewards, *e.g.*, binary accuracy reward. Although effective, GRPO training also suffers from the overthinking problem. To this end, we propose to improve the GRPO by leveraging our *InDiv* score as an extra reward, and denote this method as `H-GRPO`. The implementation details of H-GRPO can be found in Appendix A. Intuitively, by encouraging LRMs to generate diverse and concise reasoning paths, `H-GRPO` can effectively alleviate overthinking and result in better performance. It is noteworthy that the calculation of *InDiv* scores is fast and would not lead to much training latency. We apply our `H-GRPO` to reinforce the $\mathcal{M}_0$ models using the $\mathcal{D}$ dataset, and report the results of Qwen2.5 family models in Table 4.

Table 4: **Performance comparison between Qwen2.5 models using different GRPO algorithms**.

| Methods | Qwen2.5-1.5B | | Qwen2.5-3B | | Qwen2.5-7B | | Overall | |
|---|---|---|---|---|---|---|---|---|
| | MedQA | GSM8K | MedQA | GSM8K | MedQA | GSM8K | Accuracy | Tokens |
| GRPO | 46.34 | 73.08 | 57.03 | 82.49 | 66.46 | 89.61 | 69.17 | 974 |
| + Long2Short | 46.50 | 72.63 | 58.21 | 82.26 | 66.14 | 91.13 | $69.48_{\uparrow 0.31}$ | $\mathbf{668}_{\downarrow 31.4\%}$ |
| + CosFn | 47.96 | 71.72 | 55.22 | 82.93 | 63.71 | 89.31 | $68.48_{\downarrow 0.69}$ | $966_{\downarrow 0.9\%}$ |
| `H-GRPO` (Ours) | **48.15** | **74.60** | **58.98** | **83.09** | **68.03** | **91.43** | $\mathbf{70.71}_{\uparrow 1.54}$ | $710_{\downarrow 27.1\%}$ |

For comparison, we also employ two widely-used baseline methods: *Long2Short* (Team et al., 2025) and *CosFn* (Yang et al., 2025c), which address overthinking by using length-oriented reward functions. As seen, compared to the vanilla GRPO, all improved methods achieve better reasoning efficiency, indicating the validity of extra rewards. However, both length-oriented methods would cause a decrease in reasoning accuracy, *e.g.*, 1.53% average performance drops in Qwen2.5-7B on MedQA. This indicates that overly emphasizing length reduction might hinder LRMs' deep reasoning and lead to suboptimal results. Conversely, by optimizing the intermediate reasoning process, our `H-GRPO` can smoothly reduce repetitive and redundant thinking and thus achieve better performance.

**Evaluation on more reasoning benchmarks.**   To verify the generality of our `HSIR`, we additionally evaluate it on more reasoning tasks. Specifically, we use the AI2 Reasoning Challenge's challenge set (ARC) (Clark et al., 2018) for scientific reasoning, CommonsenseQA (CSQA) (Talmor et al., 2019) for commonsense reasoning, and StrategyQA (StraQA) (Geva et al., 2021) for multi-hop reasoning. Similar to the settings in §4, DeepSeek-R1 is used to collect the seed data for each task. All data and training details are provided in Appendix B. Table 5 shows the comparative results of Qwen2.5 models. As seen, our `HSIR-SFT` consistently outperforms the other counterparts by a clear margin. More specifically, compared to the base model, `HSIR-SFT` brings +5.41% average performance gains and reduces 23.3% average inference tokens. These results can prove the generality of `HSIR`.

Table 5: **Comparison results on more reasoning benchmarks**. Notably, we perform the self-improvement SFT training for one iteration in this experiment.

| Method | Qwen2.5-1.5B | | | Qwen2.5-3B | | | Qwen2.5-7B | | | Overall | |
|---|---|---|---|---|---|---|---|---|---|---|---|
| | ARC | StraQA | CSQA | ARC | StraQA | CSQA | ARC | StraQA | CSQA | Accuracy | Tokens |
| SFT-Initial | 68.33 | 59.10 | 60.03 | 79.91 | 66.67 | 71.91 | 86.43 | 72.05 | 73.63 | 70.90 | 1298 |
| SFT-Oracle | 69.38 | 60.84 | 74.20 | 80.66 | 69.29 | 79.69 | 87.55 | 74.38 | 82.47 | $75.38_{\uparrow 4.48}$ | $1223_{\downarrow 5.8\%}$ |
| STaR | 68.58 | 61.28 | 62.74 | 81.20 | 68.85 | 73.05 | 87.98 | 71.76 | 74.86 | $72.26_{\uparrow 1.36}$ | $1232_{\downarrow 5.1\%}$ |
| ReST$^{EM}$ | 69.78 | 62.30 | 68.14 | 83.26 | 68.59 | 77.64 | 89.18 | 72.36 | 79.19 | $74.49_{\uparrow 3.59}$ | $1148_{\downarrow 11.6\%}$ |
| `HSIR-SFT` | **72.45** | **64.63** | **72.48** | **83.92** | **69.59** | **78.54** | **89.87** | **73.65** | **81.65** | $\mathbf{76.31}_{\uparrow 5.41}$ | $996_{\downarrow 23.3\%}$ |

**Model Generalization.**   Here, we further investigate the ability of self-improved LRMs to generalize to out-of-distribution (OOD) tasks. Specifically, for models trained on MedQA, we evaluate their performance on the Medbullets (4-option) (Chen et al., 2025) and MedXpertQA (Zuo et al., 2025). While for models trained on GSM8K, we evaluate on the MATH (Hendrycks et al., 2021) and AMC2023 (Mathematical Association of America, 2023). We illustrate the OOD results of Qwen2.5-7B models using different self-training methods in Figure 4. From it, we can observe that: (1) Compared to iterative SFT, self-improvement with iterative DPO training can generally result in better OOD performance, similar to the finding of Wu et al. (2025). This is consistent with the wisdom that DPO can improve OOD generalization (Kirk et al., 2024). (2) Both `HSIR-SFT` and `HSIR-DPO` can achieve consistently better OOD results against the baseline methods. We attribute it to the *VeriExit* sampling strategy in `HSIR` as it can collect more diverse solutions for generalized self-training. These results confirm our motivation to mitigate the data imbalance problem.

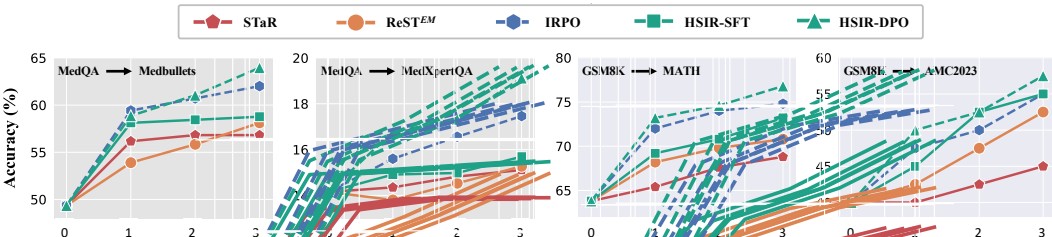

Figure 4: **Comparison of OOD results** between Qwen2.5-7B models trained with different iterative self-improvement methods. The x-axis denotes the index of self-improvement training iteration.

## 5 RELATED WORK

Recently, post-training the LLMs with explicit reasoning paths via SFT or preference learning algorithms has shown remarkable potential to unleash their reasoning capabilities (Li et al., 2025; Plaat et al., 2024; Wen et al., 2025). However, these methods are highly dependent on extensive, high-quality reasoning trajectories. Notably, although the RLVR paradigm can also enhance models' reasoning performance without relying on reasoning trajectories (Guo et al., 2025), cold-start training with these trajectories can improve training efficiency and yield higher performance (Yang et al., 2025c). This also underscores the importance of explicit reasoning trajectories. Besides obtaining these trajectories from human experts, a common alternative way is to distill them from a larger proprietary model, which is still costly and time-consuming (Peng et al., 2025).

To address the above issue, recent literature introduces the "self-improvement" paradigm, where models improve themselves using self-generated data without any external supervision (Zelikman et al., 2022; Yuan et al., 2023; Huang et al., 2023; Gulcehre et al., 2023; Wang et al., 2024; Hosseini et al., 2024; Wu et al., 2025; Huang et al., 2025a; Song et al., 2025). However, we reveal that these self-improvement methods usually suffer from data imbalance and overthinking (§2.2). Some prior studies also recognize these problems and attempt to address them by allocating more trials to difficult queries (Tong et al., 2024; Ding et al., 2025; Koh et al., 2025) or designing length-oriented reward functions to penalize too long solutions (Team et al., 2025; Munkhbat et al., 2025). For instance, AdaSTaR (Koh et al., 2025) proposes an adaptive sampling strategy to ensure data balance by prioritizing under-trained examples. While effective, it overlooks the reuse of prior failed solutions and requires a larger inference budget. Moreover, current length-oriented methods may lead to performance degradation due to excessive emphasis on length reduction (Dai et al., 2025).

Different from prior studies, we propose two simple-yet-effective approaches to address these problems efficiently. Specifically, instead of solely allocating more trials to difficult queries, our proposed *VeriExit* strategy attempts to reuse partial correct reasoning steps from previous failed solutions to improve the sampling efficiency. Notably, this technology bears some resemblance to prior early-exit decoding methods (Rahmath P et al., 2024; Yang et al., 2025b), but the idea of reusing previous failed solutions for efficient data synthesis is innovative. To alleviate overthinking, we introduce the *InDiv* score that leverages the internal state of LRMs as a signal to encourage models' diverse and concise reasoning, rather than simply using a length penalty. To the best of our knowledge, our *InDiv* is one of the first works that use the internal states of LRMs to guide the concise reasoning during self-improvement training.

## 6 CONCLUSION

In this paper, we reveal and address the limitations of self-improvement post-training in LRMs. Through a series of preliminary analyses, we find that the self-improvement of LRMs usually suffers from data imbalance and overthinking in the complex reasoning scenarios. To address these limitations, we propose HSIR, which effectively harnesses self-improvement in LRMs via two simple-yet-effective approaches: *VeriExit* sampling strategy and *InDiv* metric. Extensive results show that HSIR consistently and significantly improves the reasoning performance and efficiency across all model sizes and architectures. Moreover, we also expand our method to the RLVR training paradigm and propose H-GRPO that improves the GRPO by leveraging the *InDiv* scores as an extra reward. Comparative results with two widely-used GRPO algorithms prove the superiority of H-GRPO.

## Ethics and Reproducibility Statements

**Ethics.** We take ethical considerations very seriously and strictly adhere to the ICLR Ethics Policy. This paper proposes a new self-improvement training framework to improve the reasoning performance and efficiency of LRMs. It aims to unleash LRMs' internal reasoning capabilities, instead of encouraging them to learn privacy knowledge that may cause an ethical problem. Moreover, all base models, training and evaluation datasets used in this paper are publicly available and have been widely adopted by researchers. Thus, we believe that this research will not pose ethical issues.

**Reproducibility.** In this paper, we discuss the detailed experimental setup, such as training hyper-parameters and statistical descriptions in Appendix B. More importantly, ***we have provided our code and data in the Supplementary Material*** to help reproduce our experimental results.

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

---

**Algorithm 1** Reasoning Diversity Metric

---

1: **Input:** self-generated reasoning trajectory $\hat{r}_i^k = [\hat{r}_{i,1}^k, \ldots, \hat{r}_{i,L}^k]$, similarity threshold $\tau_{\text{sim}}$
2: **Output:** reasoning diversity score $\rho_i^k$
3: Initialize Empty Unfiltered Set $\mathcal{U}$
4: **for** Each reasoning step $\hat{r}_{i,l}^k \in \hat{r}_i^k$ **do**
5:     Obtaining the sentence embedding $emb(\hat{r}_{i,l}^k)$ using the BGE-m3 model
6:     // $Cos(emb(\hat{r}_{i,l}^k), \mathcal{U})$ denotes the cosine distance between $emb(\hat{r}_{i,l}^k)$ and its nearest neighbor in $\mathcal{U}$
7:     **if** $Cos(emb(\hat{r}_{i,l}^k), \mathcal{U}) < \tau_{\text{sim}}$ **then**
8:         $\mathcal{U} \leftarrow \mathcal{U} \cup \hat{r}_{i,l}^k$
9:     **else**
10:        Continue
11:     **end if**
12: **end for**
13: **Return:** $\rho_i^k = \frac{|\mathcal{U}|}{|[\hat{r}_{i,1}^k, \ldots, \hat{r}_{i,L}^k]|}$

---

## A   IMPLEMENTATION OF H-GRPO

**Background of GRPO and RLVR.** Group Relative Policy Optimization (GRPO) (Shao et al., 2024) is a popular RL algorithm, which is widely used in the current popular RLVR training paradigm. Formally, let $\mathcal{M}_{\theta_{\text{ref}}}$ and $\mathcal{M}_{\theta_{\text{new}}}$ denote the reference model and current policy model, GRPO samples a group of solutions $\{(\tilde{r}_i^g, \tilde{y}_i^g)\}_{g=1}^G$ for each query $x_i \in \mathcal{D}$, where $G$ denotes the number of solutions in a group. For ease of description, we simplify the solution $(\tilde{r}_i^g, \tilde{y}_i^g)$ as $a_i^g$. Then, we can optimize the $\mathcal{M}_{\theta_{\text{new}}}$ by maximizing the GRPO objective. Notably, inspired by Yu et al. (2025) who use a token-level policy gradient loss to address the unhealthy increase in response length problem of the vanilla GRPO method, we employ an improved token-level GRPO objective function as:

$$\mathcal{J}_{GRPO}(\theta) = \mathbb{E}[x_i \in \mathcal{D}, \{a_i^g\}_{g=1}^G \sim \mathcal{M}_{\theta_{\text{ref}}}(\cdot|x_i)]$$

$$\frac{1}{\sum_{g=1}^G |a_i^g|} \sum_{g=1}^G \sum_{o=1}^{|a_i^g|} \Big( \min\left(z_o(a_i^g|x_i)A_g, \text{clip}\left(z_o(a_i^g|x_i), 1-\epsilon, 1+\epsilon\right)A_g\right) - \gamma\mathbb{D}_{KL}\left(\mathcal{M}_{\theta_{\text{new}}}||\mathcal{M}_{\theta_{\text{ref}}}\right)\Big), \tag{6}$$

$$z_o(a_i^g|x_i) = \frac{\mathcal{M}_{\theta_{\text{new}}}(a_{i,o}^g|x_i, a_{i,<o}^g)}{\mathcal{M}_{\theta_{\text{ref}}}(a_{i,o}^g|x_i, a_{i,<o}^g)}, \tag{7}$$

where $\epsilon$ and $\gamma$ are hyper-parameters set to 0.2 and 0.04, respectively, $\mathbb{D}_{KL}$ is a KL penalty term, and $A_g$ is the advantage computed as follows:

$$A_g = \frac{R_g - \text{mean}(\{R_1, R_2, \cdots, R_G\})}{std(\{R_1, R_2, \cdots, R_G\})}, \tag{8}$$

where $R_g$ denotes the outcome reward of $g$-th ($g \in [1, G]$) solution in the group. In the reasoning tasks that contain clear and verifiable answers, *e.g.*, mathematical reasoning, the reward $R_g$ mainly consists of two types of rewards:

- **Accuracy reward** $R^{accuracy}$: It evaluates whether the solution is correct, *i.e.*, $\mathbb{I}(\tilde{y}_i^g, y_i) = 1$.

- **Format reward** $R^{format}$: It evaluates whether the defined tags are present in the final solution, *i.e.*, '<think>' and '</think>', '<answer>' and '</answer>' tags.

**Reward of H-GRPO.** To alleviate the overthinking problem in the vanilla GRPO, we propose H-GRPO, which improves the GRPO by leveraging our *InDiv* scores as an extra reward. Specifically, for each solution in a group, we calculate its *InDiv* score as Eq. 5, and further normalize the score as:

$$R_g^{InDiv} = \frac{\mathbf{InDiv}^g}{\max(\{\mathbf{InDiv}^1, \cdots, \mathbf{InDiv}^G\})}, \tag{9}$$

where $\max(\cdot)$ denotes the maximum *InDiv* scores in a group. The final reward for H-GRPO is the combination of all rewards:

$$R_g^{all} = R_g^{accuracy} + R_g^{format} + \omega \cdot R_g^{InDiv}, \tag{10}$$

where $\omega$ is a coefficient to control the weight of $R_g^{InDiv}$, which is set to 0.2 in our experiments.

---

**Algorithm 2** Self-improvement Training with `HSIR`

---

1: **Input:** base model $\mathcal{M}_{base}$, seed data $\mathcal{S} = \{(x_i, r_i, y_i)\}_{i=1}^N$, unlabeled dataset $\mathcal{D} = \{(x_i, y_i)\}_{i=1}^M$
2: **Output:** self-improved model $\mathcal{M}_T$
3: Fine-tune $\mathcal{M}_{base}$ on $\mathcal{S}$ to get initial reasoning model $\mathcal{M}_0$
4: **for** $t \in [1, T]$ **do**
5:      # Self-generation
6:      Obtain $K$ solutions $\{(\hat{r}_i^k, \hat{y}_i^k)\}_{k=1}^K$ generated by $\mathcal{M}_{t-1}$ for each $x_i \in \mathcal{D}$
7:      Verify the correctness of self-generated solutions, and split them into two groups:
         $\hat{\mathcal{D}}_t^{correct} = \{(x_i, \hat{r}_i^k, \hat{y}_i^k) \,|\, x_i \in \mathcal{D}; k \in [1, K]; \mathbb{I}(\hat{y}_i^k, y_i) = 1\}$
         $\hat{\mathcal{D}}_t^{wrong} = \{(x_i, \hat{r}_i^k, \hat{y}_i^k) \,|\, x_i \in \mathcal{D}; k \in [1, K]; \mathbb{I}(\hat{y}_i^k, y_i) = 0\}$
8:
9:      # *VeriExit* sampling process
10:      **for** Each sample $(x_i, \hat{r}_i^k, \hat{y}_i^k) \in \hat{\mathcal{D}}_t^{wrong}$ **do**
11:          **for** Each reasoning step $\hat{r}_{i,l}^k \in [\hat{r}_{i,1}^k, \ldots, \hat{r}_{i,L}^k]$ **do**
12:              **if** $\hat{r}_{i,l}^k$ arrives at $y_i$ **then**
13:                  Obtain a new query "$x_i + [\hat{r}_{i,1}^k, \ldots, \hat{r}_{i,l}^k] + $ `\n\n</think>\n<answer>\n`"
14:                  Feed the new query into $\mathcal{M}_{t-1}$ to resample $J$ answers $\{\hat{y}_i^{k,j}\}_{j=1}^J$
15:                  **Break**
16:              **else**
17:                  **Continue**
18:              **end if**
19:          **end for**
20:      **end for**
21:      Build a new dataset $\hat{\mathcal{D}}_t^{VeriExit} = \{(x_i, \hat{r}_{i,1\ldots l}^k, \hat{y}_i^{k,j}) \,|\, \mathbb{I}(\hat{y}_i^{k,j}, y_i) = 1; k \in [1, K]; j \in [1, J]\}$
22:      Merge $\hat{\mathcal{D}}_t^{VeriExit}$ into $\hat{\mathcal{D}}_t^{correct}$ to obtain the dataset with all correct solutions
23:
24:      # Calculate the *InDiv* score
25:      **for** Each query $x_i \in \hat{\mathcal{D}}_t^{correct}$ **do**
26:          Calculate the *InDiv* score $\mathbf{InDiv}_i^k$ for each correct solution $(\hat{r}_i^k, \hat{y}_i^k)_{k=1}^K$ as Eq. 5
27:          Get the regularized *InDiv* score $\overline{\mathbf{InDiv}}_i^k = \frac{\mathbf{InDiv}_i^k - \text{mean}(\{\mathbf{InDiv}_i^1, \cdots, \mathbf{InDiv}_i^K\})}{\text{std}(\{\mathbf{InDiv}_i^1, \cdots, \mathbf{InDiv}_i^K\})}$ for $k$-th solution
28:          Update the dataset $\hat{\mathcal{D}}_t^{correct}$ by filtering the undesired solution with $\overline{\mathbf{InDiv}}_i^k < \tau$
29:      **end for**
30:
31:      # SFT Training
32:      Fine-tune $\mathcal{M}_{base}$ with $\mathcal{L}_{\text{SFT}}$ in Eq. 1 on the combination of $\mathcal{S}$ and $\hat{\mathcal{D}}_t^{correct}$
33:      # or DPO Training
34:      Obtain a pairwise dataset $\hat{\mathcal{D}}_t^{\text{pairs}} = \{(x_i, \hat{r}_i^{k_w}, \hat{y}_i^{k_w}), (x_i, \hat{r}_i^{k_l}, \hat{y}_i^{k_l}) \,|\, x_i \in \hat{\mathcal{D}}_t; k_w, k_l \in [1, K]\}$,
         where $(\hat{r}_i^{k_w}, \hat{y}_i^{k_w}) \sim \hat{\mathcal{D}}_t^{correct}$ and $(\hat{r}_i^{k_l}, \hat{y}_i^{k_l}) \sim \hat{\mathcal{D}}_t^{wrong}$
35:      Continually train $\mathcal{M}_{t-1}$ with $\mathcal{L}_{\text{DPO+NLL}}$ in Eq. 2 on $\hat{\mathcal{D}}_t^{\text{pairs}}$
36: **end for**

---

**More analyses of Table 4.** We present the results of Qwen2.5 models trained with different GRPO methods in Table 4. Some readers may wonder why the output length of the tuned model significantly decreases after GRPO training, compared to the initial SFT, *i.e.*, from 1,540 to 987 average tokens. We conjecture that there are two main reasons. On the one hand, the token-level loss function used in the improved GRPO algorithm (Eq. 6) can effectively alleviate the abnormal increase in response length (Yu et al., 2025). On the other hand, according to the public experimental record[2] of ms-swift[3], during the GRPO training, the solution length initially decreases and then increases, indicating that the model changed its reasoning manner. Since the GRPO training is computationally expensive, we do not train the models for very long steps. That is, the training of our models may still be in the stage of decreasing output length. Despite all this, our `H-GRPO` can further reduce the inference overhead, while achieving better reasoning performance. These results can prove the superiority of `H-GRPO`, and we believe that it has great potential to perform better after longer GRPO training. Notably, the aim of this experiment is not to propose a new state-of-the-art GRPO method, but

---

[2]https://swift.readthedocs.io/en/latest/BestPractices/GRPO.html#grpo-training-experiment-record
[3]https://github.com/modelscope/ms-swift

rather to examine whether our proposed *InDiv* method can be incorporated into GRPO to effectively mitigate the over-thinking problem and address the performance degradation induced by existing length-oriented rewards. Therefore, we only compare our `H-GRPO` with two representative GRPO algorithms that rely on length-oriented rewards. In future work, we plan to further investigate how *VeriExit* and *InDiv* can be jointly incorporated into GRPO to achieve greater improvements in both training efficiency and reasoning performance.

# B MORE EXPERIMENTAL DETAILS

## B.1 DATASET DETAILS

In this work, we evaluate the trained models on several representative and challenging reasoning benchmarks. Here, we introduce the descriptions of these tasks. Specifically,

- **MedQA**: MedQA (Jin et al., 2021) is a challenging medical question-answering task, which consists of questions and corresponding 4-option or 5-option answers in the style of the US Medical License Exam (USMLE). Since the original MedQA training set does not contain the reasoning trajectories, we prompt the DeepSeek-R1 to generate the reasoning data. The prompt is shown in Table 6, and Table 7 presents a case of distilled reasoning trajectories. For in-distribution evaluation, we follow prior works (Chen et al., 2023) and use the 4-option MedQA with 1,273 samples as the test set.

- **GSM8K**: GSM8K (Cobbe et al., 2021) is a widely-used mathematical reasoning task, which contains 8.5K high-quality grade school math word problems. Since the original GSM8K does not contain any reasoning trajectories, we alternatively use the GSM8K version[4] released by CAMEL (Li et al., 2023). Notably, the dataset is also distilled from the DeepSeek-R1. Table 8 presents a case of distilled GSM8K training data. For in-distribution evaluation, we directly use the original GSM8K with 1.32K test samples.

- **Medbullets**: Medbullets (Chen et al., 2025) comprises 308 difficult USMLE Step 2&3 style medical questions collected from real-world conversations. Each question is paired with a case description and multiple answer choices. In our work, we use the 4-option Medbullets as the OOD test set of LRMs trained on MedQA.

- **MATH**: MATH (Hendrycks et al., 2021) comprises 500 problems spanning five core mathematical domains: algebra, combinatorics, geometry, number theory, and precalculus. Each problem is designed to test the multi-step and complex reasoning abilities of LRMs, requiring more than simple calculation or knowledge recall. In our experiments, we use this challenging dataset to evaluate the OOD performance of models trained on GSM8K.

- **MedXpertQA**: MedXpertQA (Zuo et al., 2025) contains 4,460 high-difficulty medical questions spanning 17 specialties and 11 body systems. It includes two subsets, MedXpertQA Text for text medical evaluation and MedXpertQA MM for multimodal medical evaluation. We use the MedXpertQA Text as the OOD test set of LRMs trained on MedQA.

- **AMC2023**: AMC2023 (Mathematical Association of America, 2023) consists of 40 challenging mathematical problems from American Mathematics Competitions, which is widely used to evaluate the complex reasoning performance of LRMs. We use the public test set[5] to evaluate the OOD performance of models trained on GSM8K.

- **ARC**: AI2 Reasoning Challenge's challenge set (ARC) (Clark et al., 2018) is a scientific reasoning dataset that contains 1.12K multiple-choice science QA training samples and 1.17K test samples. Similar to MedQA, we prompt the DeepSeek-R1 to generate the reasoning steps for the training samples, and randomly select 500 samples as the seed data, using the remaining training samples as unlabeled data. The trained models are evaluated on the ARC test set.

- **CommonsenseQA**: CommonsenseQA (CSQA) (Talmor et al., 2019) is a multiple-choice question-answering dataset that requires diverse types of commonsense knowledge to predict the correct answers. It contains 12,102 questions, each with one correct answer and four distractors. Similarly, For the 9.74K training samples, we distill reasoning steps from DeepSeek-R1 and randomly select 1K samples as seed data. The trained models are evaluated on the 1.14K test samples.

---

[4]https://huggingface.co/datasets/camel-ai/gsm8k_distilled
[5]https://huggingface.co/datasets/zwhe99/amc23

- **StrategyQA**: StrategyQA (StraQA) (Geva et al., 2021) is an implicit multi-hop reasoning bench-mark, which contains 1.6K training samples and 687 test samples. After distilling the reasoning steps from DeepSeek-R1 for the training samples, we randomly select 500 training samples as the seed data. The trained models are directly evaluated on the test samples.

## B.2 TRAINING AND EVALUATION DETAILS

In the SFT phase, we fine-tune each model with a batch size of 8 and a peak learning rate of 1e-5, except 2e-6 for 7B/8B models. In the DPO phase, the batch size is set to 16, and the peak learning rates for smaller (1.5B/3B) models and larger(7B/8B) models are set to 1e-6 and 2e-7, respectively. In both training settings, the warm-up ratio is 0.1, and the maximum tokenizer length is 2,048. All models are trained with full parameter fine-tuning for 3 epochs. We use the popular `LLaMA-Factory`[6] as the training codebase.

For the RLVR training, we implement the GRPO with the `open-r1`[7] framework. The rollout size for each query is set to 8, the learning rate is set to 1e-6, and the batch size is set to 14. Since the GRPO training is time-consuming and costly, we only train the models for 300 steps. For the implementation of our `H-GRPO`, the coefficient $\omega$ to control the weight of $R^{InDiv}$ is set to 0.2. All experiments are conducted on 8 NVIDIA A800 (80GB) GPUs.

During evaluation, we sample the model responses by using the greedy decoding method, *i.e.*, temperature set to 0 for reproducibility. The maximum output length is set 4,096. All models are evaluated in a zero-shot manner. We extract the final answer from the model response according to the given format. If no valid answer is extracted, the response is considered incorrect.

Table 6: **Prompts used for distilling reasoning paths** for MedQA from DeepSeek-R1.

---

**System Prompt**:
You are a helpful medical AI Assistant. Your task is to select one correct answer from A to D for the medical question. Please respond in the following format: '##Explanation:\n[explanation]\n\n##Answer:\n[option]'

---

**User Prompt**:
For the following medical question, select one correct answer from A to D.
*Question*: {Question}
*Options*:
A. {Option-A}
B. {Option-B}
C. {Option-C}
D. {Option-D}

---

Table 7: **An example of seed data in MedQA** distilled from DeepSeek-R1.

---

**System Prompt**:

You are a helpful medical AI Assistant that provides well-reasoned and detailed responses for medical questions. You first think about the reasoning process as an internal monologue and then provide the user with the concise and accurate answer. Respond in the following format: <think>\n...\n</think>\n<answer>\n...\n</answer>

---

**Instruction**:

For the following medical question, select one correct answer from A to D.

---

[6]https://github.com/hiyouga/LLaMA-Factory
[7]https://github.com/huggingface/open-r1

*Question*: A 55-year-old woman comes to the physician because of fevers for 2 weeks. She works as a nurse and recently returned from a charity work trip to India, where she worked in a medically-underserved rural community. A tuberculin skin test 3 months ago prior to her trip showed an induration of 3 mm. Physical examination is unremarkable. An x-ray of the chest shows right-sided hilar lymphadenopathy. A sputum culture shows acid-fast bacilli. Which of the following immunologic processes most likely occurred first?

*Options*:

A. Formation of a nodular tubercle in the lung

B. Replication of bacteria within alveolar macrophages

C. Production of interferon-gamma by T-helper cells

D. Migration of T-helper cells to the lungs

Please respond in the following format: '##Explanation:\n[explanation]\n\n##Answer:\n[option]'

**Output**:

<think> *#reasoning trajectory*

Okay, let's tackle this question step by step. The patient is a 55-year-old nurse who recently returned from India and has symptoms of TB. Let's break down the information given.

She had a tuberculin skin test (TST) three months ago with 3 mm induration, which is considered negative (since the cutoff for healthcare workers is usually 10 mm). Now, she has a positive sputum culture for acid-fast bacilli, which points to active tuberculosis. The chest X-ray shows hilar lymphadenopathy, which is common in primary TB, especially in the right side.

The question is asking about the first immunologic process that occurred. The options are about the sequence of immune responses when someone is infected with Mycobacterium tuberculosis.

So, the initial step when TB bacteria enter the body is inhalation. The bacilli reach the alveoli, where they are taken up by alveolar macrophages. However, M. tuberculosis has mechanisms to survive inside these macrophages. The bacteria can replicate within the macrophages until the macrophages are overwhelmed and lyse, releasing more bacteria. This replication within macrophages is an early step before the immune system starts to respond more effectively.

After that, the adaptive immune response kicks in. T-helper cells (CD4+ T cells) migrate to the site of infection. These cells release cytokines like interferon-gamma, which activate macrophages to become more effective at killing the bacteria. The activated macrophages form granulomas (tubercles) to contain the infection. The formation of the granuloma is part of the body's attempt to wall off the infection.

So the sequence would be: bacteria enter macrophages and replicate (B), then T-cells migrate (D), produce interferon-gamma (C), leading to granuloma formation (A).

The question asks which happened first. So the first step after exposure would be the replication within macrophages. The TST was negative before her trip, suggesting she wasn't previously infected. Now she's positive, so this is a new infection. The initial infection's first immunologic process is the bacteria replicating in macrophages before the immune response starts.

</think>

<answer> *#answer*

##Explanation:

The patient's recent exposure and subsequent active tuberculosis indicate a new infection. The initial immunologic process begins when Mycobacterium tuberculosis is inhaled and phagocytosed by alveolar macrophages. The bacteria survive and replicate within these macrophages (option B), which occurs before the adaptive immune response. Subsequent steps include T-helper cell migration to the lungs (D), interferon-gamma production (C), and granuloma formation (A). The replication within macrophages is the earliest event in the immunologic cascade.

##Answer:

B

</answer>

Table 8: **An example of seed data in GSM8K** distilled from DeepSeek-R1.

**System Prompt**:

You are a helpful AI Assistant that provides well-reasoned and detailed responses for the math word problems. You first think about the reasoning process as an internal monologue and then provide the user with the concise and accurate answer. The final answer should be provided in the \boxed{}.

**Instruction**:

John starts at an elevation of 400 feet. He travels downward at a rate of 10 feet down per minute for 5 minutes. What is his elevation now?

**Output**:

<think> *#reasoning trajectory*

Okay, so I need to figure out John's elevation after he travels downward for 5 minutes at a rate of 10 feet per minute. Let me start by understanding the problem. John starts at 400 feet, and he's moving downward. Moving downward would decrease his elevation, right? So, he's going from higher elevation to lower.

First, I should note his starting point: 400 feet. Then, he's going down at 10 feet per minute for 5 minutes. So, I think I can calculate the total distance he descended by multiplying the rate by the time. That makes sense because if you go 10 feet every minute, then in 5 minutes, that's 10 * 5 = 50 feet. Wait, yes, so he would have gone down 50 feet in total.

Since he's moving downward, his new elevation will be the original elevation minus that descent. So, subtracting 50 feet from 400 feet. Let me do that calculation: 400 - 50. That gives 350 feet. Hmm, so his elevation after 5 minutes should be 350 feet.

Wait, let me check that again. Multiplication first: 10 feet per minute times 5 minutes is indeed 50 feet. Then subtract that from the starting elevation. Yeah, 400 minus 50 is 350. I think that's correct. Maybe I should visualize this. Imagine he's on a hill at 400 feet. Each minute, he goes down 10 feet, so after 1 minute, he's at 390, then 380, 370, 360, 350 after 5 minutes. Each step is 10 feet lower. So yes, five steps would be 50 feet total.

Is there any chance I might have mixed up the direction? Like thinking he's ascending? But the problem clearly states he's traveling downward. So subtracting is the right operation. Okay, I feel confident that this is right. Let me just go through the steps again to confirm. Starting elevation 400. Rate: 10 ft/min downward. Time: 5 min. Total descent is rate * time = 10*5=50 ft. New elevation: 400 - 50 = 350 ft. Yeah, that checks out. I think that's the correct answer.

</think>

*#answer*

John starts at an elevation of **400 feet**.

To find his elevation after descending:

1. **Calculate total descent**: Rate = 10 feet/minute × 5 minutes = **50 feet**.

2. **Subtract the descent from the starting elevation**: 400 feet – 50 feet = **350 feet**.

**Verification**: Each minute, his elevation decreases by 10 feet:

- After 1 minute: 400 – 10 = 390 feet

- After 2 minutes: 390 – 10 = 380 feet

- Continuing this pattern for 5 minutes leads to **350 feet**.

**Final Answer**:

\boxed{350}

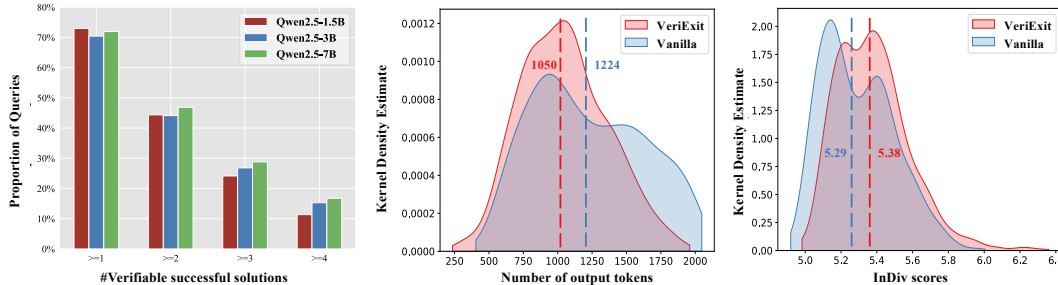

Figure 5: **Left**: Coverage of verifiable successful solutions. The x-axis denotes the number of verifiable successful solutions in a query. **Middle**: Distribution of the number of output tokens of correct self-generated solutions. **Right**: Distribution of *InDiv* scores of correct self-generated solutions. In the middle and right sub-figures, we use the initial SFT Qwen2.5-1.5B models.

## C  MORE EXPERIMENTS AND ANALYSES

### C.1  ANALYSIS OF *VeriExit* SAMPLING STRATEGY

**Reliability of *VeriExit*.**  In HSIR, we propose *VeriExit* to efficiently collect more successful solutions for difficult queries by reusing partial correct reasoning steps of previous failed solutions. Here, to verify the reliability of this idea, we calculate the coverage of these verifiable successful solutions, which contain intermediate reasoning steps that arrived at the correct answer. Figure 5 (**Left**) illustrates the coverage of these solutions in the most difficult queries of MedQA. Specifically, we refer to the queries that did not obtain any correct solutions during the $K$-times ($K = 10$ in our experiments) self-generation processes as the most difficult ones. The x-axis denotes the number of verifiable successful solutions in a query, and the y-axis denotes the proportion of queries. As seen, among all Qwen2.5 family models, nearly 70% of these most difficult queries have at least one verifiable successful solution, and more than 10% of these queries have 4 or more verifiable successful solutions. We believe that in the simpler queries, there is a larger proportion of previous failed solutions that can be converted into verifiable successful solutions by *VeriExit*. These results can demonstrate the reliability of our *VeriExit* Sampling Strategy.

Moreover, we compare the solutions resampled by *VeriExit* and the previous correct solutions. In practice, we measure the length and *InDiv* scores of solutions on the difficult MedQA queries (obtaining four correct solutions during the previous self-generation) generated by initial SFT Qwen2.5-1.5B models, and visualize the distributions in Figure 5 (**Middle** and **Right**). As seen, compared to the previous correct solutions generated by the vanilla sampling strategy, our *VeriExit* can sample shorter and concise reasoning trajectories. We attribute it to the self-truncation and early-exit processes of *VeriExit*, which can skip the redundant and repetitive reasoning steps. This can also explain why the "-w/ VeriExit" method in Table 3 can improve reasoning efficiency against the ReST$^{EM}$.

**More *VeriExit* variants.**  The core of our *VeriExit* is to determine whether the reasoning step has reached the ground-truth answer. There are several methods to achieve this goal. Specifically, the simplest approach is to directly match the text between the reasoning output and the ground-truth answer. If the reasoning output explicitly mentions the answer, *e.g.*, "answer is $\{y_i\}$", we can assume that it arrives at the ground-truth answer. Beyond this simple heuristic, inspired by SelfCheckGPT (Manakul et al., 2023), we can further use two more sophisticated approaches: **NLI-based** and **prompt-based**. In the **NLI-based** *VeriExit*, we use an external Natural Language Inference (NLI) model to judge the relationships between the reasoning output and answer. The NLI model can determine whether the reasoning output entails the answer according to the similarity of sentence representations, *i.e.*, $\mathbb{I}(\text{NLI}(\hat{r}_{i,l}^k, y_i) = \text{entailment})$. In practice, we use the powerful DeBERTa-v3-large-mnli[8] as the NLI model. In the **prompt-based** *VeriExit*, we query the current $\mathcal{M}_{t-1}$ to assess whether the reasoning output and answer convey the same meaning by using the following prompt: "*You are a semantic-equivalence classifier. Your only goal is to decide whether the two input sentences convey the same meaning. Please direct output the answer following the*

---

[8]https://huggingface.co/MoritzLaurer/DeBERTa-v3-large-mnli-fever-anli-ling-wanli

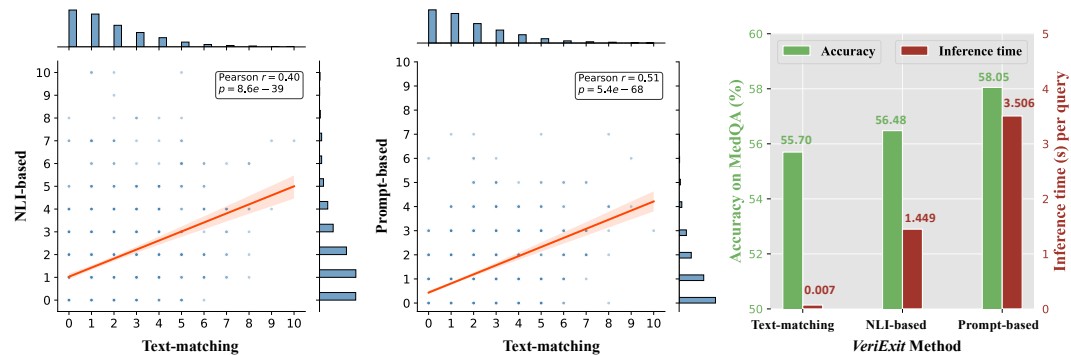

Figure 6: **Left**: Correlation between text-matching and NLI-based *VeriExit* methods. **Middle**: Correlation between text-matching and prompt-based *VeriExit* methods. **Right**: Performance and efficiency comparisons of `HSIR-SFT` variants equipped with different *VeriExit* methods. In the left and middle sub-figures, the axises denote the number of verifiable successful solutions per query. Qwen2.5-3B model is used in this experiment and all models are self-improved for one iteration.

*format: '##Answer: [YES|NO|UNCLEAR]'"*. Notably, for both methods, we convert the answer into a full sentence using the template "the answer is $y_i$" to ensure consistent semantic comparison.

To evaluate different *VeriExit* strategies, we measure the coverage of verifiable successful solutions on the most difficult queries of MedQA using each strategy. Using the text-matching method as the baseline, we illustrate the correlation between text-matching and NLI-based/prompt-based methods in Figure 6 (**Left**) and (**Middle**), respectively. In this experiment, we use the Qwen2.5-3B as the base model. The results show that text-matching *VeriExit* correlates well with both alternative methods, with Pearson Correlation Coefficients exceeding 0.4 and p-values below 0.05, indicating that all *VeriExit* variants produce largely consistent predictions. Further, we replace the *VeriExit* strategy in the `HSIR-SFT` framework, and compare the performance and efficiency of `HSIR-SFT` variants in Figure 6 (**Right**). The findings are as follows: 1) both NLI-based and prompt-based methods achieve better reasoning performance, as they can more accurately identify the correct reasoning steps during *VeriExit*; 2) although effective, these methods incur more additional inference overhead. Therefore, for simplicity and efficiency, we use the text-matching method in our work by default.

## C.2 ANALYSIS OF *InDiv* SCORE

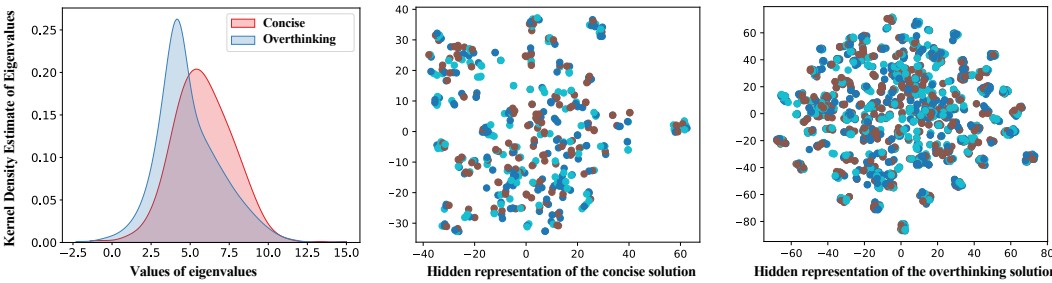

Figure 7: **Left**: Distribution of eigenvalues of hidden representation in the concise and overthinking solutions. **Middle**: t-SNE visualizations of hidden representations in the concise solution. **Right**: t-SNE visualizations of hidden representations in the overthinking solution. Here, we use the initial SFT Qwen2.5-1.5B as the test model. The concise and overthinking solutions are from Table 15.

**Correlation between *InDiv* scores and semantic entropy.**    Here, we investigate the correlation between our *InDiv* scores and the semantic entropy of hidden representations **H**. First, we introduce the **Matrix-Based Entropy** (Giraldo et al., 2014; Skean et al., 2023; 2025), which is a famous information-theoretic quantity. For the cross-covariance matrix $\mathbf{\Sigma}_i^k = \mathbf{H}_i^{k\top} \cdot \mathbf{J}_d \cdot \mathbf{H}_i^k$ and its

eigenvalues $\{\lambda_{i,u}^k\}_{u=1}^m$, the matrix-based entropy of order $\alpha > 0$ is:

$$\mathbf{S}_{\alpha,i}^k(\mathbf{H}_i^k) = \frac{1}{1-\alpha} \log \Big[ \sum_{u=1}^m (\frac{\lambda_{i,u}^k}{\mathrm{Tr}(\mathbf{H}_i^k)})^\alpha \Big], \tag{11}$$

where $\mathrm{Tr}(\cdot)$ denotes the trace operator obtained from the sum of $\alpha$-power of each eigenvalues (Horn & Johnson, 2012). When $\alpha \to 1$, the entropy $\mathbf{S}_{\alpha,i}^k(\mathbf{H}_i^k)$ corresponds to the Shannon's entropy of hidden representations. Intuitively, if the eigenvalues of $\mathbf{H}$ are in a uniform distribution, the entropy will be higher, indicating that $\mathbf{H}$ contains more diverse features (Skean et al., 2025). Conversely, if the eigenvalues collapse to a small area, the entropy will be smaller. To verify it, we compare the distributions of eigenvalues between the concise and overthinking solutions identified by our *InDiv* scores. Specifically, we directly use the solutions in Table 15, and illustrate their eigenvalue distributions in Figure 7 (**Left**). It can be found that the eigenvalue distributions of the concise solution are more uniform than those of the overthinking solution. To have a closer look, we directly visualize the hidden representations of both solutions. Figure 7 (**Middle**) and (**Right**) show the t-SNE results of the concise and overthinking solutions, respectively. We can observe that the distribution of hidden representations of the concise solution is more diverse and uniform, while that of the overthinking solution is more similar and concentrated. Overall, these results indicate that a higher *InDiv* score usually refers to a higher semantic entropy of hidden representations, which effectively proves why our *InDiv* score can help identify overthinking solutions.

**Impact of layer depth for calculating *InDiv* scores.** As mentioned in §3.2, we use the hidden representations from the middle layer of $\mathcal{M}_{t-1}$ to calculate the *InDiv* scores. Here, we investigate the impact of different layer depths by comparing the performance of Qwen2.5-1.5B models trained with different `HSIR-SFT` configurations on MedQA. Specifically, since the Qwen2.5-1.5B contains 28 layers, we vary the layer used for calculating *InDiv* scores across {5, 10, 15, 20, 25} and illustrate the comparative results in Figure 8. For reference, we also include the results of SFT-Initial and ReST$^{EM}$ methods. All models are self-improved for one iteration. As seen, `HSIR-SFT` with varied layer depth can consistently outperform the other baseline methods, indicating that `HSIR` is relatively robust to the choice of layer. Moreover, when using the middle layer (*i.e.*, 15-th layer), `HSIR-SFT`

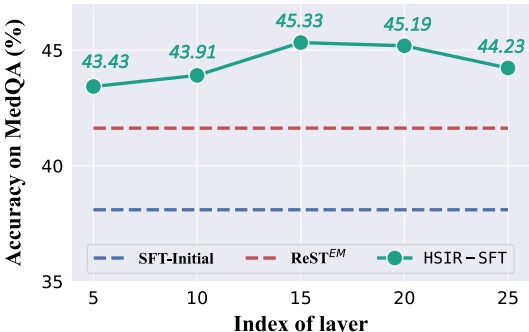

Figure 8: **Analysis of different layer depths for calculating *InDiv* scores**. Notably, we use the Qwen2.5-1.5B (with a total of 28 layers) as the test model. All models are trained for one self-improvement iteration.

achieves the best performance. We conjecture that the middle layer encodes richer and more useful semantic information (Skean et al., 2025; Azaria & Mitchell, 2023; Liu et al., 2019), thus resulting in more accurate *InDiv* scores. Based on these observations, we choose to adopt the middle layer of $\mathcal{M}_{t-1}$ for calculating *InDiv* scores in this work.

**More ablation study in *InDiv*.** There are two important strategies in our *InDiv*: attention-aware weighting mechanism and *InDiv* normalization. The former is to obtain more fine-grained intrinsic diversity, while the latter is to regularize the *InDiv* score for more flexible data filtering. To evaluate their contributions, we compare our full `HSIR-SFT` with two variants: 1) "-w/o attention-aware", which removes the attention-aware weighting mechanism, i.e., $Atten_{i,u}^k = \frac{1}{m}$ in Eq. 5; 2) "-w/o normal-

Table 9: **Analysis of important components in *InDiv***. All models are self-improved for one iteration.

| Method | Qwen2.5-1.5B | | Qwen2.5-3B | |
|---|---|---|---|---|
| | MedQA | GSM8K | MedQA | GSM8K |
| SFT-Initial | 38.10 | 63.99 | 49.02 | 77.18 |
| **HSIR-SFT** | **45.33** | **71.72** | **55.70** | **86.13** |
| -w/o attention-aware | 45.20 | 71.34 | 55.38 | 85.75 |
| -w/o normalization | 43.52 | 71.27 | 54.13 | 85.60 |

ization", which directly uses the $\mathbf{InDiv}_i^k$ in Eq. 5 as the final score and filters the solutions with lower scores within each query. Notably, The filtering ratio is adjusted to ensure that both variants

use approximately the same amount of training data as the original `HSIR-SFT`. Table 9 presents the results, showing that removing either strategy leads to performance degradation. These findings consistently demonstrate the effectiveness of both strategies.

### C.3 PARAMETER ANALYSIS

**Effect of $\tau$.** The threshold $\tau$, used to filter the overthinking solutions, is an important hyperparameter in our `HSIR`. In this study, we analyze its influence by evaluating the performance with different $\tau$ values, spanning from -0.75 to 0.75. Figure 9 illustrates the comparative results of Qwen2.5-3B models trained with `HSIR-SFT` on MedQA and GSM8K. For reference, we also report the results without data filtering as the baseline. As seen, compared to the baseline, `HSIR` with suitable $\tau$ can generally achieve better performance, showing the effectiveness of using `InDiv` scores to filter overthinking solutions. However, too large $\tau$ (*i.e.*, 0.75) would lead to performance degradation, as many helpful training samples might be ignored. `HSIR` performs best with $\tau = -0.5$, thus leaving as our default experimental settings.

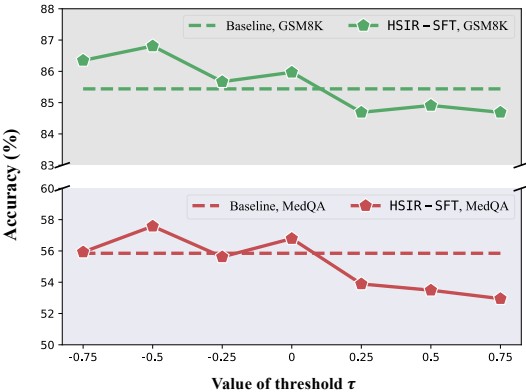

Figure 9: **Analysis of threshold $\tau$.** "Baseline" means that we do not filter the overthinking solutions, *i.e.*, removing the *InDiv* in `HSIR`.

**Effect of $J$.** The hyperparameter $J$, which is used to control the sample count of recycled solutions during *VeriExit*, is also important for our `HSIR`. Here, to investigate its impact, we evaluate our `HSIR-SFT` with different $J$ values ranging from 1 to 7. The comparative results of Qwen2.5-1.5B on MedQA are illustrated in Figure 10. For reference, we also include the results of SFT-Initial and ReST$^{EM}$. All models are self-improved for one iteration. From these results, we find that: 1) When $J$ is too small (*e.g.*, $J = 1$), our *VeriExit* struggles to sample enough correct solutions, limiting the effectiveness of `HSIR-SFT`. 2) When $J$ is too large (*e.g.*, $J = 7$), many sampled solutions share similar prefix reasoning steps, reducing the diversity of

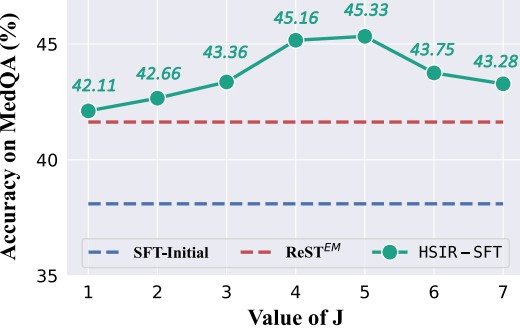

Figure 10: **Analysis of sampling count $J$.** We use the Qwen2.5-1.5B as the test model. All models are trained for one self-improvement iteration.

training data and leading to sub-optimal performance. 3) Across all $J$ values, our `HSIR-SFT` consistently outperforms the baseline methods, proving its robustness. Notably, in the case of $J = 5$, `HSIR-SFT` achieves the best performance. Thus, we use it as the default setting in this work.

### C.4 IMPACT OF SEED DATA

**Seed data from QWQ-32B.** As mentioned in §2.1, we first fine-tune the base model $\mathcal{M}_{base}$ on the seed dataset $\mathcal{S}$ to make it have basic long-CoT reasoning abilities. Intuitively, high-quality seed data can improve the basic reasoning ability of LRMs and boost the effectiveness of self-improvement training. To verify it, we replace the seed data used in our main experiments with that distilled from QWQ-32B[9]. Taking the MedQA as an example, we fine-tune the Qwen2.5 family models on the seed data distilled from QWQ-32B and self-generated pseudo-labeled data for one iteration, using different self-improvement training methods. Table 10 reports the average accuracy and number of output tokens of all tuned models, from which we find that: (1) Compared to the seed data distilled from DeepSeek-R1, the seed data generated by QWQ-32B performs differently in different models. For the smaller models (*i.e.*, Qwen2.5-1.5B), the seed data from QWQ-32B brings more performance

---

[9] https://qwenlm.github.io/blog/qwq-32b-preview/

gains. Conversely, for the larger 7B model, it leads to worse results. We conjecture that there is a large capacity gap between Qwen2.5-1.5B and DeepSeek-R1, while using a smaller QWQ-32B as a teacher can achieve a smooth knowledge transfer (Mirzadeh et al., 2020; Zhang et al., 2023). However, for the Qwen2.5-7B, a smaller teacher model might struggle to provide sufficient knowledge, thus leading to suboptimal performance. (2) When using the seed data generated by QWQ-32B, our HSIR can still outperform the other baseline methods and achieve better reasoning performance and efficiency. These results demonstrate the universality and robustness of HSIR.

Table 10: **Performance comparison of the seed data distilled from different LRMs**. We evaluate the Qwen2.5 models fine-tuned with different self-improvement SFT methods for one iteration.

| Methods | \|Train\| Avg. | Qwen2.5-1.5B | | Qwen2.5-3B | | Qwen2.5-7B | | Overall | |
|---|---|---|---|---|---|---|---|---|---|
| | | MedQA | Tokens | MedQA | Tokens | MedQA | Tokens | Accuracy | Tokens |
| *Seed data distilled from DeepSeek-R1* | | | | | | | | | |
| SFT-Initial | 1.0K | 38.10 | 1,779 | 49.02 | 1,644 | 62.45 | 1,428 | 49.86 | 1,617 |
| *Seed data distilled from QWQ-32B* | | | | | | | | | |
| SFT-Initial | 1.0K | 40.22 | 1,677 | 50.04 | 1,569 | 61.12 | 1,607 | 50.46 | 1,618 |
| SFT-Oracle | 8.0K | 43.99 | 1,781 | 60.25 | 1,451 | 69.13 | 1,419 | 57.79 $_{\uparrow 7.33}$ | 1,550 $_{\downarrow 4.2\%}$ |
| STaR | 4.9K | 39.35 | 1,576 | 49.25 | 1,500 | 61.82 | 1,541 | 50.14 $_{\downarrow 0.32}$ | 1,539 $_{\downarrow 4.9\%}$ |
| ReST$^{EM}$ | 37.9K | 44.38 | 1,431 | 54.77 | 1,307 | 65.43 | 1,401 | 54.86 $_{\uparrow 4.40}$ | 1,380 $_{\downarrow 14.7\%}$ |
| **HSIR–SFT** | 29.6K | **46.58** | **1,167** | **55.70** | **1,192** | **66.93** | **1,266** | **56.40** $_{\uparrow 5.94}$ | **1,208** $_{\downarrow 25.3\%}$ |

**Is the seed data from a frontier model necessary?** In the above experiments, we empirically find that initial LRMs trained with high-quality seed data can effectively self-improve via our HSIR. Here, we conduct more in-depth experiments to investigate whether the seed data distilled from a frontier model is necessary. Specifically, we focus on two types of seed data: 1) **low-quality seed data**, which is distilled from a smaller and weaker LRM; 2) **self-distilled seed data**, which is generated by the model itself via an in-context learning approach. Using the Qwen2.5-3B as the testbed, we obtain low-quality

Table 11: **Analysis of different seed data**. We report the MedQA results of Qwen2.5-3B trained with different self-improvement SFT methods for one iteration.

| Method | Source of seed data | | | |
|---|---|---|---|---|
| | DeepSeek | QWQ | Qwen2.5-1.5B | Self-distilled |
| SFT-Initial | 49.02 | 50.04 | 42.50 | 48.70 |
| STaR | 49.25 | 49.25 | 43.36 | 47.53 |
| ReST$^{EM}$ | 55.22 | 54.77 | 47.18 | 49.80 |
| **HSIR–SFT** | **55.70** | **55.70** | **48.00** | **50.17** |

seed data from the SFT-Initial Qwen2.5-1.5B. For the self-distilled seed data, we randomly select three examples distilled from DeepSeek-R1 as few-shot demonstrations and use them to guide the base Qwen2.5-3B for generating the seed data. Notably, for all methods, we use the same queries $x_i$ and sample a correct solution for each query, ensuring the same number of training samples across seed data types. Qwen2.5-3B model is first fine-tuned on different seed data, and then self-improved with various SFT approaches for one iteration. Table 11 presents the comparative results of different Qwen2.5-3B models on MedQA. From it, we observe that: 1) the quality of seed data is critical, as initial SFT on low-quality seed data significantly degrades performance; 2) across all seed data types, our HSIR consistently brings performance gains, further validating its effectiveness. In general, while it is feasible for an LLM to self-distill seed data and subsequently self-improve using HSIR, leveraging more high-quality seed data from stronger frontier models allows HSIR to realize its full potential and achieve better performance.

## C.5 WHEN GROUND-TRUTH ANSWER ARE UNAVAILABLE

Following many prior studies (Zelikman et al., 2022; Yuan et al., 2023; Wang et al., 2024; Pang et al., 2024), we assume that the ground-truth answers of unlabeled dataset $\mathcal{D}$ are available in this work. Some readers may wonder how our HSIR performs when ground-truth answers are unavailable in some scenarios. Actually, in this setting, we can follow Huang et al. (2023) and use the majority-voting answer among multiple candidate solutions as a pseudo answer, *i.e.*, $\tilde{y}_i = \arg\max_{\hat{y}_i^j} \sum_{k=1}^{K} \mathbb{I}(\hat{y}_i^j = \hat{y}_i^k)$. Here, the $\tilde{y}_i$ is denoted as the self-consistency pseudo label. Although the $\tilde{y}_i$ may be incorrect, we can still apply our HSIR to improve the LRMs as described in §3. To verify its effectiveness, we evaluate it on several Qwen2.5 models and report the results in Table 12. For reference, we use the

ReST$^{EM}$ as the baseline method, and also include the results based on ground-truth answers. As seen, using the self-consistency pseudo labels indeed yields slightly sub-optimal results compared to ground-truth answers. Nevertheless, our `HSIR-SFT` method still achieves substantial improvements, with an average gain of +6.14%, proving that `HSIR` remains effective even in unlabeled scenarios.

Table 12: **Performance comparison of Qwen2.5 models on MedQA and GSM8K benchmarks**. Notably, "SC →" refer to using the majority-voting answer among multiple candidate solutions of SFT-Initial models as the pseudo labels of $\mathcal{D}$. All models are self-improved for one iteration.

| Methods | Qwen2.5-1.5B | | Qwen2.5-3B | | Qwen2.5-7B | | Overall | |
|---|---|---|---|---|---|---|---|---|
| | MedQA | GSM8K | MedQA | GSM8K | MedQA | GSM8K | Accuracy | Tokens |
| SFT-Initial | 38.10 | 63.99 | 49.02 | 77.18 | 62.45 | 83.93 | 62.45 | 1,536 |
| *Using ground-truth answers* | | | | | | | | |
| ReST$^{EM}$ | 41.63 | 69.75 | 55.22 | 83.95 | 64.18 | 88.17 | 67.15$_{\uparrow 4.70}$ | 1,268$_{\downarrow 17.5\%}$ |
| HSIR-SFT | **45.33** | **71.72** | **55.70** | **86.13** | **67.32** | **88.78** | **69.16**$_{\uparrow 6.71}$ | 1,075$_{\downarrow 30.0\%}$ |
| *Using self-consistency pseudo labels* | | | | | | | | |
| SC → ReST$^{EM}$ | 40.46 | 70.36 | 51.69 | 83.70 | 64.57 | 87.04 | 66.30$_{\uparrow 3.85}$ | 1,322$_{\downarrow 13.9\%}$ |
| SC → HSIR-SFT | **44.78** | **71.72** | **54.99** | **84.69** | **65.91** | **89.46** | **68.59**$_{\downarrow 6.14}$ | 1,108$_{\downarrow 27.8\%}$ |

## C.6 RESULTS IN HIGH-RESOURCE SCENARIOS

In our work, we assume that only a small amount of seed data is available. Some readers may wonder whether our `HSIR` method remains effective in high-resource scenarios, where sufficient seed data is provided. To verify it, we use all training samples distilled from DeepSeek-R1 as the seed data to initially fine-tune the base model, *i.e.*, using the SFT-Oracle in Table 1 as the $\mathcal{M}_0$. Table 13 presents the results of Qwen2.5-1.5B and Qwen2.5-3B models on MedQA. For reference, we also report the results of ReST$^{EM}$ and IRPO as baselines.

Table 13: **Results in high-resource scenarios**. The SFT-Oracle is used as the initial $\mathcal{M}_0$, which is trained with self-improvement methods for one iteration.

| Method | Qwen2.5-1.5B | | Qwen2.5-3B | |
|---|---|---|---|---|
| | MedQA | Tokens | MedQA | Tokens |
| SFT-Oracle | 46.58 | 1,678 | 58.68 | 1,448 |
| *Using SFT-Oracle as the initial model $\mathcal{M}_0$* | | | | |
| ReST$^{EM}$ | 47.21$_{\uparrow 0.63}$ | 1,423$_{\downarrow 15.2\%}$ | 61.81$_{\uparrow 3.13}$ | 1,243$_{\downarrow 14.2\%}$ |
| HSIR-SFT | 50.90$_{\uparrow 4.32}$ | 1,156$_{\downarrow 31.1\%}$ | 63.71$_{\uparrow 5.03}$ | 1,116$_{\downarrow 22.9\%}$ |
| IRPO | 46.35$_{\downarrow 0.23}$ | 1,683$_{\uparrow 0.3\%}$ | 59.46$_{\uparrow 0.78}$ | 1,585$_{\uparrow 9.5\%}$ |
| **HSIR-DPO** | 53.49$_{\uparrow 6.91}$ | 1,047$_{\downarrow 37.6\%}$ | 66.61$_{\uparrow 7.93}$ | 1,116$_{\downarrow 22.9\%}$ |

All models are self-improved for one iteration. From these results, we find that both `HSIS-SFT` and `HSIS-DPO` methods can effectively improve the performance and inference efficiency of SFT-Oracle models. Specifically, with the help of `HSIS-DPO`, Qwen2.5-3B model achieves +7.93% performance gain on MedQA. These findings demonstrate that our `HSIS` has great potential to enhance the self-improvement capabilities of fully-trained and powerful LRMs.

## C.7 EFFICIENCY OF HSIR

Some readers may be concerned about the efficiency of our `HSIR` method, as it requires additional forward passes of LRMs. Actually, during the *VeriExit* sampling phase, we only sample the final answer without regenerating intermediate reasoning trajectories, making it much faster than simply allocating more trials to failed queries. In our preliminary experiments, we found that the *VeriExit* sampling can be completed in an average of one hour on 8 NVIDIA A800 (80GB) GPUs, which is about 1/4 of the time required for generating full reasoning trajectories. On the other hand, to obtain the *InDiv* scores, the query and its solution are fed into the model in a teacher-forcing manner, requiring only a single forward pass and introducing minimal latency. In practice, during `HSIR-SFT` and `HSIR-DPO` training, obtaining hidden states for each token introduces some computational overhead, but this can be completed in about half an hour on 8 NVIDIA A800 (80GB) GPUs. Moreover, during the `H-GRPO` training, we can reuse the hidden representation obtained by the reference model without extra forward passes. The actual computation of the InDiv score itself is lightweight, involving only simple vector operations that take a few seconds. More importantly, owing to the data filtering process of `HSIR`, the training budget can be significantly reduced, *e.g.*, from 53.6K to 38.8K during fine-tuning Qwen2.5-1.5B at the last iteration. In general, the inference latency of `HSIR` is tolerable against its performance gains.

## C.8 MORE SELF-IMPROVEMENT ITERATIONS

Due to limited computation resources, we set the maximum self-improvement iterations $T$ to 3 in the main experiments. Here, to further investigate whether additional iterations can improve performance, we extended the maximum training iterations $T$ from 3 to 5, and compare the MedQA accuracy of Qwen2.5-1.5B models trained with different self-improvement SFT training methods across the iterations. Figure 11 illustrates the comparative results, from which we observe that: 1) With the increase of self-improvement training iterations, both STaR and ReST$^{EM}$ exhibit a trend where performance initially improves but then declines, which is similar to the findings of Ding et al. (2025). This may be due to overfitting on easy-to-learn samples. Conversely, by mitigating the data imbalance problem, our HSIR can collect more challenging samples and achieve continuous performance improvements. 2) As self-improvement training progresses, the performance gains of HSIR tend to be smaller, indicating the existence of an upper-bound for self-improvement training.

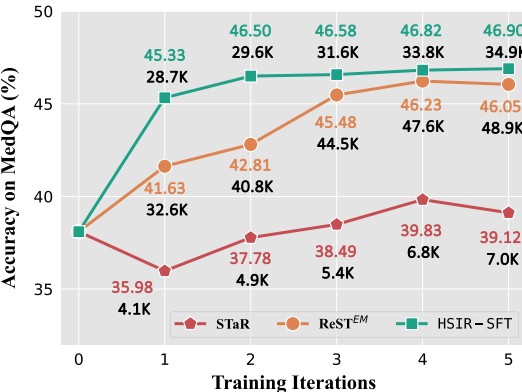

Figure 11: **Results of Qwen2.5-1.5B models training for more self-improvement iterations**. Here, we report both test accuracy and the number of training samples on MedQA.

## C.9 COMPATIBILITY WITH SELF-CONSISTENCY

The goal of our work is to propose a self-improvement training framework that unlocks the internal long-CoT reasoning capabilities of LRMs, rather than to optimize inference. Therefore, in the main experiments, we do not compare HSIR with inference-time methods, such as Self-Consistency (SC) (Wang et al., 2023). Nevertheless, given that SC is widely used to enhance LRM reasoning performance, we include it in this experiment. Specifically, during inference, we sample five solutions from the model and select the majority-vote answer as the prediction. Figure 12 shows the comparative results of Qwen2.5 models on MedQA, where "-w/ SC" means using the SC method. As seen, increasing the test-time compute through SC improves the reasoning performance of SFT-Initial models effectively. However, it still underperforms our HSIR-SFT method, even if HSIR-SFT only samples a single solution during inference. More encouragingly, combining the HSIR-SFT and SC methods consistently yields further performance improvements. For instance, for the Qwen2.5-3B model, with the help of SC, HSIR-SFT achieves a 12.25% performance gain over the SFT-Initial model. These results prove the compatibility of our HSIR with inference-time SC method.

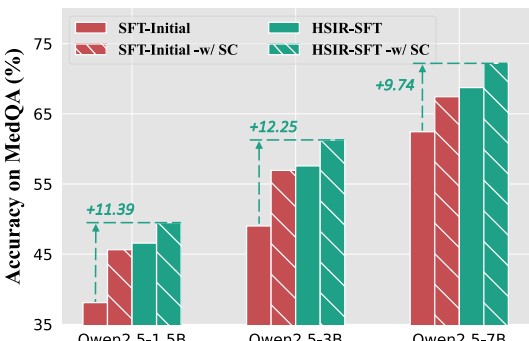

Figure 12: **Performance comparison between with and without the self-consistency method**. Notably, we report the results of HSIR-SFT after three self-improvement iterations.

## C.10 CASE STUDY

To have a closer look, we provide some cases in this part. Specifically, Table 14 shows the comparison of self-generated solutions between before and after using our *VeriExit* sampling strategy. We show the self-generated results of Qwen2.5-1.5B on the MedQA task. As seen, in the previous failed solutions, the model indeed arrived at the correct answer during intermediate reasoning steps, yet ultimately failed to produce the accurate outcome due to reasoning deviation. Conversely, by using our verify-then-exit strategy, we can convert it into a collect solution. Table 15 shows the comparison of *InDiv* scores between concise and overthinking self-generated solutions. Obviously, *InDiv* scores

can reflect the redundancy and repetitiveness of reasoning steps generated by Qwen2.5-1.5B on the MedQA task. We observe that the overthinking solution indeed obtains a much lower *InDiv* score. These case studies support the intuition and soundness of our proposed methods.

Table 16 and Table 17 show the comparison of reasoning results of Qwen2.5-7B models before and after training using HSIR on MedQA and GSM8K test sets, respectively. It can be found that, with the help of HSIR, the model can reason more accurately and concisely, thus resulting in better performance. These case studies prove the effectiveness of HSIR.

Table 14: **An example of converting the failed solution into the correct one via the *VeriExit* strategy**. In this case from MedQA, the ground-truth answer is "D". As seen, the failed solution arrived at the correct answer during intermediate reasoning steps (highlighted in green), but diverted into an incorrect one due to overly backtracking (highlighted in red).

---

**Previous Failed Solution**:

\<think\>

Okay, let's try to work through this question.

. . .

So, if the options include allopurinol (option C), then that's the answer. But the options given here are D. Xanthine oxidase inhibitor. Wait, but xanthine oxidase inhibitors are allopurinol and other drugs like febuxostat. So, allopurinol is a xanthine oxidase inhibitor. So, if the answer is C, then allopurinol. But the question is asking for the long-term treatment. So, if the options don't include allopurinol, then perhaps the answer is D. Wait, I think the answer is C. Allopurinol. So, the correct answer would be C.

. . .

##Answer:

C

\</answer\>

---

**New Solution Generated by *VeriExit***:

\<think\>

Okay, let's try to work through this question.

. . .

So, if the options include allopurinol (option C), then that's the answer. But the options given here are D. Xanthine oxidase inhibitor. Wait, but xanthine oxidase inhibitors are allopurinol and other drugs like febuxostat. So, allopurinol is a xanthine oxidase inhibitor. So, if the answer is C, then allopurinol. But the question is asking for the long-term treatment. So, if the options don't include allopurinol, then perhaps the answer is D.

\</think\> *(verify then exit)*

\<answer\>

. . .

##Answer:

D

\</answer\>

---

Table 15: **Comparison of *InDiv* scores between concise and overthinking solutions in a single query**. Here, we use the case of MedQA. Compared to the concise solution, the overthinking solution contains more repetitive reasoning steps (highlighted in orange).

---

**Concise Solution**:

\<think\>

Okay, let's try to figure this out. So the question is about malnutrition in children and which amino acid is needed for them. The symptoms described are Kwashiorkor, which is protein deficiency. The options are A. Alanine, B. Tyrosine, C. Arginine, D. Methionine.

First, I remember that kwashiorkor is typically due to inadequate protein intake. The main issue is the lack of essential amino acids, especially the ones that the body can't make on its own. The nine essential amino acids are tryptophan, threonine, histidine, lysine, isoleucine, leucine, methionine, phenylalanine, and valine.

Looking at the options, Methionine (D) is one of those essential amino acids. Alanine (A) is a non-essential one, so maybe not needed here. Tyrosine (B) is an amino that's non-essential and involved in protein synthesis but not for this presentation. Arginine (C) is also a non-essential one and more associated with metabolic pathways like making creatine or proteins, not really for this symptom. So the answer should be D. Methionine.

</think>

. . .

</answer>

*InDiv* Score: `5.99`

**Overthinking Solution**:

<think>

Okay, let's tackle this question.

. . .

So the answer would be D. Methionine. Wait, but I thought Tyrosine is an essential amino acid . . .

So perhaps the answer is D. Methionine, because in some cases, the body uses up all the available EAA's . . .

So the answer is D. Methionine. Because in kwashiorkor, without proper nutrition, the body cannot utilize other amino acids. So supplementing with Methionine would help . . .

Therefore, the correct answer is D. Methionine. Because even if the diet has methionine, the body can't utilize it. . . .

Alternatively, maybe the answer is Tyrosine. But I think not. So the answer is D.

</think>

. . .

</answer>

*InDiv* Score: `5.01`

Table 16: **Comparison of MedQA reasoning results of Qwen2.5-7B models before and after training using `HSIR`**. The ground-truth answer of this case is "D".

**Instruction**:

For the following medical question, select one correct answer from A to D.

*Question*: A 25-year-old African American man presents to his primary care provider for routine blood work. He is a well-known case of sickle cell disease treated with hydroxyurea and tramadol as needed for pain. He does not smoke or drink alcohol. The patient asks if his disease can adversely affect his kidneys because his friend was recently diagnosed with end-stage renal failure. Which of the following is a renal complication of sickle cell disease?

*Options*:

A. Membranous glomerulonephritis

B. Proximal renal tubular acidosis

C. Impaired urine diluting ability

D. Impaired urine concentration ability

**Output of SFT-Intial Model**:

<think>

Okay, let's tackle this question. So, the patient is a 25-year-old African American man with sickle cell disease. He's on hydroxyurea and tramadol for pain. His friend has end-stage renal failure, and he's asking if his disease can affect his kidneys. The question is about a renal complication of sickle cell disease. The options are A to D.

First, I need to recall the common renal complications associated with sickle cell disease. Sickle cell disease can lead to various kidney issues. Let me think. The kidneys are affected because of the hemolytic anemia and the high levels of hemoglobin S, which can cause microangiopathic hemolysis.

Membranous glomerulonephritis (option A) is a type of nephrotic syndrome. I remember that in sickle cell disease, membranous nephropathy is a known complication. It's an immune complex-mediated process, leading to proteinuria and nephrotic syndrome. So that's a possibility.

Proximal renal tubular acidosis (option B) is a genetic disorder where the kidneys can't excrete acids properly. But I don't think that's linked to sickle cell. PRSA is more about a defect in the renal tubules' ability to handle acid, which isn't directly related to sickle cell's hemolysis.

Impaired urine diluting ability (option C) would be nephrogenic diabetes insipidus. That's when the kidneys can't concentrate urine due to a defect in response to ADH. Again, not directly linked to sickle cell. Maybe from other causes like certain drugs or genetic disorders.

Impaired urine concentration ability (option D) is nephrogenic diabetes insipidus as well. So this is the same as option C. So D is not the right answer here.

So the options are between A and C. Wait, the question is about a renal complication. Sickle cell can cause various issues. Let me think again. The patient's friend had end-stage renal failure, which could be due to chronic kidney disease from sickle cell. The options given are specific conditions.

Membranous glomerulonephritis (A) is a type of glomerular disease leading to nephrotic syndrome. In sickle cell, membranous nephropathy is a recognized complication, often presenting with nephrotic syndrome. So A is a possible answer.

Impaired urine diluting ability (C) would be nephrogenic DI, which is unrelated. So the correct answer here is A. Membranous glomerulonephritis. </think>

<answer>

. . .

##Answer:

A

</answer>

---

**Output of `HSIR-DPO` Model**:

<think>

Okay, let's tackle this question. So, the patient is a 25-year-old African American man with sickle cell disease. He's on hydroxyurea and tramadol for pain. His friend had end-stage renal failure, and the patient wants to know if his disease can affect his kidneys. The options are about different renal complications.

First, I need to recall the common renal issues associated with sickle cell disease. Sickle cell disease can cause various complications, including kidney problems. I remember that sickle cell can lead to different glomerular diseases. For example, sickle cell nephropathy, which is a type of glomerulonephritis. But looking at the options, option A is membranous glomerulonephritis. I think that's more commonly associated with conditions like lupus or hepatitis B, but maybe I'm mixing up. Alternatively, maybe it's a rare complication.

Wait, another possibility is that sickle cell can cause tubular issues. The options include proximal renal tubular acidosis (option B). I know that sickle cell can lead to Fanconi syndrome, which is a form of proximal tubular dysfunction. Fanconi syndrome includes glucosuria, aminoaciduria, etc. But option B is specifically proximal RTA, which is a different condition. Proximal RTA is when the proximal tubule can't reabsorb properly, leading to loss of nutrients and acidosis. But I'm not sure if that's directly linked to sickle cell.

Alternatively, the question might be pointing towards glomerular issues. The options include impaired urine diluting ability (option C) and impaired concentration ability (option D). I remember that in sickle cell disease, there's a condition called hypernephrosis, which is an increase in kidney size due to fluid overload. This can lead to dilutional hyponatremia because the kidneys can't concentrate urine properly. So, impaired concentration ability (option D) would be the issue here. But wait, the options are C and D as separate. Wait, impaired urine diluting ability—so the kidneys can't excrete concentrated urine. Impaired concentration ability would be the opposite, right? So in sickle cell, the kidneys might have trouble concentrating urine, leading to dilutional hyponatremia. So option D would be the correct answer here.

But wait, another thought: sickle cell can cause glomerulonephritis leading to proteinuria, hematuria, etc. But the options don't mention that. The options given are specific. So, if the question is about a renal complication, maybe the answer is D.

</think>

<answer>

. . .

##Answer:

D

</answer>

---

Table 17: **Comparison of GSM8K reasoning results of Qwen2.5-7B models before and after training using `HSIR`**. The ground-truth answer of this case is "20".

---

**Instruction**:

Toby is reading a book that is 45 pages long. It averages 200 words a page. Toby can read at a rate of 300 words per minute. He has to be at the airport in 60 minutes and plans to leave as soon as he finishes the book. It takes 10 minutes to get to the airport. How many minutes early will Toby be?

---

**Output of SFT-Intial Model**:

<think>

Okay, let's see. So the problem is about Toby reading a book and figuring out how early he'll be at the airport. Hmm. Let me start by understanding each part step by step.

First, the book is 45 pages long. Each page averages 200 words. So, to find the total number of words in the book, I need to multiply the number of pages by the average words per page. That would be 45 pages times 200 words per page. Let me calculate that. 45 times 200... 45 times 200 is 9,000 words. So the book has 9,000 words in total.

Next, Toby's reading rate is 300 words per minute. So, to find out how long it takes him to read the entire book, I need to divide the total number of words by his reading rate. That's 9,000 words divided by 300 words per minute. Let me do that division. 9,000 divided by 300. Hmm, 300 times 30 is 9,000. So that's 30 minutes. So Toby will take 30 minutes to read the book.

But wait, he has to be at the airport in 60 minutes, and it takes him 10 minutes to get there. So the total time he needs to leave his house is 60 minutes (time to be at the airport) plus 10 minutes (travel time) which is 70 minutes. But he only needs 30 minutes to read the book. So if he starts reading now, he'll finish in 30 minutes. Then he has 70 - 30 = 40 minutes left before he needs to leave. Therefore, he'll be 40 minutes early.

Wait, but let me check again to make sure I didn't miss anything. Let me recap: Total words in the book: 45 pages * 200 words/page = 9,000 words. Reading time: 9,000 words / 300 words per minute = 30 minutes. Time needed to get to the airport: 10 minutes. Total time before he needs to leave: 60 minutes.

So time he has after finishing reading: 60 + 10 - 30 = 40 minutes. Therefore, he's 40 minutes early. That seems right. I don't think I made any mistakes here. The key steps were calculating the total words, then the reading time, and then subtracting that from the total time he has before he needs to leave. Yeah, that makes sense.

</think>

. . .

**Final Answer**: \boxed{40}

---

**Output of `HSIR-DPO` Model**:

<think>

Okay, let's see. Toby has a 45-page book to read. Each page averages 200 words. He reads at 300 words per minute. He has 60 minutes before he needs to be at the airport, and it takes him 10 minutes to get there. The question is how many minutes early he'll be if he reads the whole book and leaves right after. Hmm, let me break this down step by step.

First, I need to figure out how many words are in the entire book. Since there are 45 pages and each page has 200 words, I can multiply those two numbers. So 45 pages times 200 words per page. Let me calculate that. 45 times 200... 45 times 200 is 9,000 words. So the book has 9,000 words total.

Next, Toby reads at a rate of 300 words per minute. To find out how long it takes him to read the entire book, I need to divide the total number of words by his reading rate. That's 9,000 words divided by 300 words per minute. Let me do that division. 9,000 ÷ 300. Hmm, 300 times 30 is 9,000. So that's 30 minutes. So reading the book takes him 30 minutes.

But he has to get to the airport, which takes 10 minutes. So the total time from when he starts reading until he leaves for the airport is the time reading plus the time to get there. That's 30 minutes + 10 minutes = 40 minutes. Wait, but he has 60 minutes before he needs to be at the airport. So if he finishes reading and leaves 40 minutes later, he'll have 60 - 40 = 20 minutes left. That means he'll be 20 minutes early.

Let me just verify each step again. Pages times words per page: 45*200=9000. Reading time: 9000/300=30. Time to airport:10. Total time:30+10=40. Time left:60-40=20. Yep, that checks out. So the answer is 20 minutes early.

</think>

. . .

**Final Answer**: \boxed{20}

---

# D   LLM USAGE

During the paper writing, we used the proprietary LLMs as general-purpose assistants to polish some sentences. We did not prompt them to generate research ideas or search for related works, thus avoiding the risk of introducing false information. In the experiments, we instructed the DeepSeek-R1 to obtain the intermediate reasoning trajectories for the unlabeled MedQA training set. The detailed prompt is shown in Table 6. The way of distilling reasoning trajectories from proprietary LLMs is common in the community, and we believe that this behavior does not violate the policies.

