# OpenReview forum: "Better, Faster: Harnessing Self-Improvement in Large Reasoning Models"
_ICLR.cc/2026/Conference — Submitted to ICLR 2026_

### Official Review · Reviewer_tSEv · 2025-10-27

**Soundness:** 3
**Presentation:** 4
**Contribution:** 2
**Rating:** 6
**Confidence:** 4

**Summary:**

This paper tackles two limitations of self-improvement in reasoning models: data imbalance, where training data skews toward simple problems, and overthinking, where models generate redundant and repetitive reasoning steps. The authors propose HSIR, a framework featuring two core components to make models "better and faster". First, a VeriExit sampling strategy addresses data imbalance by efficiently salvaging correct intermediate reasoning from failed solutions for difficult queries . Second, an Intrinsic Diversity score, derived from an attention-aware eigenvalue analysis of the model's internal hidden states, quantifies and filters out "overthinking" solutions. HSIR is applied to both supervised fine-tuning and preference learning, and its InDiv score is also used as a reward in a new RL algorithm, H-GRPO. Experiments on medical and math reasoning tasks show HSIR boosts performance and improves efficiency by reducing inference overhead.

**Strengths:**

Writing is clear to follow. Visualizations are well designed.

It is good that this approach works on both (SFT, DPO) and RL

Diverse model sizes and model families are used for experimentation.

**Weaknesses:**

The work should mention how it compares and contrasts with “AdaSTaR: Adaptive Data Sampling for Training Self-Taught Reasoners” (NeurIPS 2025) as their method also related to:

> Through a series of preliminary analyses, we reveal two shortcomings of self-improvement in LRMs: (1) data imbalance, where most training samples are simple, but the challenging yet crucial samples are scarce; (2) overthinking, where many undesired samples with redundant and repetitive reasoning steps are used for self-training.

The ❶ Data Imbalance section in the proposed paper has been introduced in this previous work.

Unclear how this is a good set-up to analze self-improvement. I.e., T=1 has almost no self improvement?
> For the implementation of self-improvement training, the number of iterations T is set to 1, and the total sampling times K is set to 10.

The paper should clearly mention all compute cost overhead incurred by the method over existing methods.

> sentence embeddings using the BGE-m3 model

The training data scope is limited to only two: gsm8k and medqa.

Is the seed data from a frontier model necessary? What are the compute overheads incurred due to this. Does any frontier model suffice?

Are all the hyperparameters systematically set? Any sensitivity tests? It is especially unclear if T=1 is reasonable as mentioned above.

> The filter threshold τ is set to -0.5. Notably, for the post-training of Qwen2.5 models, the self-improvement iteration T is set to 3, but for the other LLMs, it is set to 1 due to limited computational resources.

Although E.4 PARAMETER ANALYSIS is available, it would be nice to double check if any more hyperparameters are newly introduced and might be sensitive. If it is sensitive it may be a pain for practitioners.

What is the unit of train budget in e.g. table 1. Is it reflective of wall-clock time as well?

In industry labs we would want to see that peak performance, since we can checkpoint and go back to this one. However, it is not very convincing to see a fixed iteration budget performance, i.e., 3. Any experiments were we can see peak val set performance comparisons?

In table 2, SFT-Oracle seems to perform consistently better. In what scenario would we not have access to this? I.e., if there is already labeled data available, what is the point of using this method? Any experiments to show orthogonality? E.g. SFT-oracle + proposed method?

The gains on GRPO seem negligibly small in Tab. 4.

**Questions:**

Why call Iterative Reasoning Preference Optimization RPO? IRPO seems like the most appropriate abbreviation.

---

> ### Author Response · Authors · 2025-11-23
> **Response to Reviewer tSEv (1/3)**
>
> We thank the reviewer for the positive and valuable comments and are glad that he/she recognizes the generality of our method, and appreciates both the clarity of the writing and the comprehensiveness of our experiments. The responses to the questions are below:
>
> > W1. The work should mention how it compares and contrasts with “AdaSTaR: Adaptive Data Sampling for Training Self-Taught Reasoners” (NeurIPS 2025) as their method also related to data imbalance problem.
>
> Thank you for pointing this out. We acknowledge that AdaSTaR also addresses the data imbalance problem, but **there are several key differences** between their work and ours:
> - **Different methods**: AdaSTaR proposes an adaptive sampling strategy to ensure data balance by prioritizing under-trained examples. Although effective, this approach focuses on allocating more trials to difficult queries and does not leverage previously failed solutions. **In contrast, our VeriExit salvages valid initial reasoning from failed attempts, providing a significant efficiency gain over costly resampling from scratch**.
> - **Different motivation**: AdaSTaR primarily targets data imbalance, whereas our work **also addresses the over-thinking problem**, which is critical in the long-CoT reasoning scenarios.
>
> We have clarified the difference between AdaSTaR and ours in the Related Work section of the revised paper.
>
> ---
>
> > W2. Unclear how this is a good set-up to analze self-improvement. I.e., T=1 has almost no self improvement?
>
> We would like to clarify that “T=1” means that **we perform the self-improvement training for one iteration**, i.e., updating the model from $\mathcal{M}_0$ to $\mathcal{M}_1$. We have clarified it in Section 2.2 of the revised paper.
>
> ---
>
> > W3. The paper should clearly mention all compute cost overhead incurred by the method over existing methods.
>
> The main computation overhead incurred by our method over the vanilla self-improvement method arises from the VeriExit and Indiv strategies.
>
> For VeriExit, we only sample the final answer without regenerating the intermediate reasoning trajectories, which is much faster than simply allocating more trials for failed queries. In our preliminary experiments, we found that the VeriExit sampling can be finished in **approximately one hour on 8 NVIDIA A800 (80GB) GPUs**, which is about 1/4 of the time required to generate full reasoning trajectories.
>
> For InDiv, the query and its solution are fed into the model in a teacher-forcing manner, requiring only a single forward pass and minimal latency. In practice, during HSIR-SFT and HSIR-DPO training, obtaining hidden states for each token introduces some computational overhead, but this can be completed in **about half an hour on 8 NVIDIA A800 (80GB) GPUs**. Moreover, during H-GRPO training, we can reuse the hidden representation obtained by the reference model without extra forward passes. **The actual computation of the InDiv score itself is lightweight, involving only simple vector operations that take a few seconds**. We have clarified these details in Appendix C7 of our revised paper.

---

> > ### Author Response · Authors · 2025-11-23
> > **Response to Reviewer tSEv (2/3)**
> >
> > > W4. The training data scope is limited to only two: gsm8k and medqa.
> >
> > To demonstrate the generality of our HSIR method, we further evaluate it on additional reasoning benchmarks: **ARC** [1] for scientific reasoning, **CommonsenseQA** [2] for commonsense reasoning, and **StrategyQA** [3] for multi-hop reasoning. The results are presented in **Q3** of the **General Response**. Overall, HSIR consistently outperforms all baseline methods across these benchmarks, confirming its broad generality. We have added these experiments in Section 4.3 of the revised paper.
> >
> > ---
> >
> > > W5. Are all the hyperparameters systematically set? Any sensitivity tests? It would be nice to double check if any more hyperparameters are newly introduced and might be sensitive.
> >
> > Actually, we had reported all relevant hyperparameters in Section 4.1 of the original paper. In particular, there are two important hyperparameters in our HSIR, i.e., threshold $\tau$ and sampling count J in VeriExit. **The effect of the threshold $\tau$ was analyzed in Appendix E4 of the original paper**. While for J, following your suggestion, **we further evaluate the sensitivity of HSIR-SFT to J by testing values from 1 to 7**. The comparative results of Qwen2.5-1.5B on MedQA are presented in the following table. For reference, we also include the results of SFT-Initial and ReST$^{EM}$. All models are self-improved for one iteration.
> >
> > | Method | MedQA |
> > |---|---|
> > | SFT-Initial | 38.1 |
> > | ReST$^{EM}$ | 41.63 |
> > | HSIR-SFT |  |
> > | J=1 | 42.11 |
> > | J=2 | 42.66 |
> > | J=3 | 43.36 |
> > | J=4 | 45.16 |
> > | J=5 | 45.33 |
> > | J=6 | 43.75 |
> > | J=7 | 43.28 |
> >
> > From these results, we find that: 1) When J is too small (e.g., J=1), our VeriExit struggles to sample enough correct solutions, limiting the effectiveness of HSIR. 2) When J is too large (e.g., J=7), there are too many similar VeriExit-sampled solutions that share the same prefix reasoning steps. These similar solutions would reduce the diversity of training data and lead to sub-optimal results. 3) **Across all J values, our HSIR consistently outperforms the baseline methods, proving its robustness**. Notably, in the case of J=5, HSIR achieves the best performance, thus leaving it as the default setting in our work. We have added this experiment to Appendix C3 of the revised paper.
> >
> > ---
> >
> > > W6. What is the unit of train budget in e.g. table 1. Is it reflective of wall-clock time as well?
> >
> > The training budget in Table 1 refers to **the average number of training samples across all models and tasks**. This metric can reflect the average wall-clock training time. We have clarified it in the revised version.
> >
> > ---
> >
> > > W7. In industry labs we would want to see that peak performance, since we can checkpoint and go back to this one. However, it is not very convincing to see a fixed iteration budget performance, i.e., 3. Any experiments were we can see peak val set performance comparisons?
> >
> > In our preliminary experiments, we empirically observed that **the final model checkpoint usually achieves the best performance**. Therefore, we report the results of the last checkpoint in each iteration as the peak performance. Additionally, *the original paper already presented performance across different training iterations* (Iterations 1, 2, and 3) in Table 1.
> >
> > To further investigate whether additional iterations can improve performance, we extended the maximum training iterations from 3 to 5 and report the comparative results of Qwen2.5-1.5B models on MedQA in the following table.
> >
> > | Iteration | STaR | ReST | HSIR-SFT |
> > |---|---|---|---|
> > | 1 | 35.98 | 41.63 | 45.33 |
> > | 2 | 37.78 | 42.81 | 46.50 |
> > | 3 | 38.49 | 45.48 | 46.58 |
> > | 4 | **39.83** | **46.23** | 46.82 |
> > | 5 | 39.12 | 46.05 | **46.90** |
> >
> > From these results, we observe that: 1) With the increase of self-improvement training iterations, both STaR and ReST$^{EM}$ exhibit a trend where performance initially improves but then declines, which is similar to the findings of [4]. This may be due to overfitting on easy-to-learn samples. Conversely, by mitigating the data imbalance problem, our HSIR can collect more challenging samples and achieve continuous performance improvements. 2) As self-improvement training progresses, the performance gains of HSIR tend to be smaller, indicating **the existence of an upper bound for self-improvement training**. We have added this experiment and analysis to Appendix C8 of the revised paper.

---

> ### Author Response · Authors · 2025-11-23
> **Response to Reviewer tSEv (3/3)**
>
> > W8. In table 2, SFT-Oracle seems to perform consistently better. In what scenario would we not have access to this? I.e., if there is already labeled data available, what is the point of using this method? Any experiments to show orthogonality? E.g. SFT-oracle + proposed method?
>
> Thanks for your valuable question. To verify **whether our HSIR method remains effective in high-resource scenarios**, we use all training samples distilled from DeepSeek-R1 as the seed data to initially fine-tune the base model, i.e., **using the SFT-Oracle as $\mathcal{M}_0$**. The following table presents the results of Qwen2.5-1.5B and Qwen2.5-3B models on MedQA. For reference, we also report the results of ReST$^{EM}$ and IRPO. All models are self-improved for one iteration.
>
> | Method | Qwen2.5-1.5B | Qwen2.5-3B |
> |---|:---:|:---:|
> | SFT-Oracle ($\mathcal{M}_0$) | 46.58 | 58.68 |
> | STaR+ | 47.21 | 61.81 |
> | HSIR-SFT | 50.9 | 63.71 |
> | IRPO | 46.35 | 59.46 |
> | HSIR-DPO | 53.49 | 66.61 |
>
> From these results, we find that both HSIS-SFT and HSIS-DPO methods can effectively improve the performance and inference efficiency of SFT-Oracle models. Specifically, with the help of HSIS-DPO, the Qwen2.5-3B model achieves +7.93% performance gain on MedQA. These results demonstrate that **our HSIS has great potential to enhance the self-improvement capabilities of fully-trained and high-performance LRMs**. We have added this experiment to Appendix C6 of our revised paper.
>
> ---
>
> > W9. The gains on GRPO seem negligibly small in Tab. 4.
>
> We would like to clarify that **the goal of H-GRPO is not to propose a new state-of-the-art GRPO method**, but rather to *examine whether our InDiv can be incorporated into GRPO to effectively mitigate the over-thinking problem* and address the performance degradation induced by existing length-oriented rewards. Therefore, we simply incorporate InDiv as an additional reward signal in the vanilla GRPO. In future work, we plan to *explore more sophisticated integrations of VeriExit and InDiv into GRPO to achieve further improvements in both efficiency and performance*. We have clarified it in Appendix A of our revised paper.
>
> ---
>
> > Q1. Why call Iterative Reasoning Preference Optimization RPO? IRPO seems like the most appropriate abbreviation.
>
> In the original paper [5], the method is abbreviated as “Iterative RPO”, and we simplified it as "RPO" in our original paper. Following your suggestion, we have updated the abbreviation to "IRPO" in the revised paper for clarity and consistency.
>
> ---
>
> ### Reference
>
> [1] Think you have solved question answering? try arc, the ai2 reasoning challenge, arxiv2018. \
> [2] Commonsenseqa: A question answering challenge targeting commonsense knowledge, NAACL2019. \
> [3] Did aristotle use a laptop? a question answering benchmark with implicit reasoning strategies, TACL2021. \
> [4] Mitigating Tail Narrowing in LLM Self-Improvement via Socratic-Guided Sampling, NAACL2025. \
> [5] Iterative reasoning preference optimization, NeurIPS2024.

---

> ### Comment · Reviewer_tSEv · 2025-11-26
>
> Many thanks to the authors for their effort, and reasonable rebuttal.
>
> w2. yes, i understand, why not larger T?
>
> w4. the new experiments are quite limited in comprehensiveness, however this is understandable as the rebuttal phase is short. Nevertheless, this seems like a major revision that needs work on.
>
> other experimental weaknesses that were mentioned also need a more comprehensive run through; this may be challenging to finish by the rebuttal phase. nevertheless, these are important points that should be covered for real-world impact.
>
> I will maintain my score 6, and confidence remains the same.

---

> ### Author Response · Authors · 2025-11-27
> **Further Response to Reviewer tSEv**
>
> Thanks for your valuable feedback and for acknowledging the reasonableness of our response. To address the concerns you still have, we provide further clarification below:
> > w2: why not larger T?
>
> We would like to clarify that the primary goal of Section 2.2 is to quickly reveal the shortcomings of vanilla self-improvement training through preliminary experiments. Moreover, due to limited computational resources, we conducted only a single iteration of self-improvement training in these experiments. While we believe this does not affect the reliability of our conclusions, we are willing to include experiments with more self-improvement iterations in subsequent phases.
>
> ---
>
> > w4: limited comprehensiveness in the new experiments
>
>
> We acknowledge that, due to time constraints, we struggle to conduct fully comprehensive experiments during the rebuttal phase. However, we make our best effort to include additional experiments on more models in this phase. The expanded experiments on the Qwen2.5-3B and Qwen2.5-7B models across these benchmarks are presented in the tables below, which further support the credibility of our conclusions. These results have been updated in **Q3** of **General Response** and incorporated into the revised manuscript.
>
> - **More results on Qwen2.5-3B:**
>
> | Method      |  ARC  | StrategyQA |  CSQA | Avg.  | Tokens |
> |-------------|:-----:|:----------:|:-----:|-------|--------|
> | SFT-Initial | 79.91 |    66.67   | 71.91 | 72.83 |  1,271 |
> | SFT-Oracle  | 80.66 |    69.29   | 79.69 | 76.55 |  1,239 |
> | STaR        | 81.20 |    68.85   | 73.05 | 74.37 |  1,226 |
> | ReST$^{EM}$        | 83.26 |    68.59   | 77.64 | 76.50 |  1,144 |
> | HSIR-SFT    | **83.92** |    **69.59**   | **78.54** | **77.35** |  **1,030** |
>
> - **More results on Qwen2.5-7B:**
>
> | Method      |  ARC  | StrategyQA |  CSQA | Avg.  | Tokens |
> |-------------|:-----:|:----------:|:-----:|-------|--------|
> | SFT-Initial | 86.43 |    72.05   | 73.63 | 77.37 |  1,177 |
> | SFT-Oracle  | 87.55 |    74.38   | 82.47 | 81.47 |  1,098 |
> | STaR        | 87.98 |    71.76   | 74.86 | 78.20 |  1,114 |
> | ReST$^{EM}$        | 89.18 |    72.36   | 79.19 | 80.24 |  1,045 |
> | HSIR-SFT    | **89.87** |    **73.65**   | **81.65** | **81.72** |   **914**  |
>
> ---
>
> Regarding your concerns about the comprehensiveness of the other experiments, we assure you that we will continue to include additional experimental results in subsequent phases to further strengthen the reliability of our conclusions.
>
> Thank you again for your feedback, and we hope our explanations address your concerns. If you have any further questions, we would be glad to continue the discussion.

---

### Official Review · Reviewer_DUEw · 2025-10-29

**Soundness:** 1
**Presentation:** 2
**Contribution:** 1
**Rating:** 2
**Confidence:** 4

**Summary:**

This paper focuses on self-improvement scenarios for llm reasoning.
The study identifies several key factors that affect a model's self-improvement capability, such as overly simplistic problems and model overthinking.

To address these issues, the paper proposes the Harnesses Self-Improvement in large Reasoning (HSIR) method:
1. verify-then-exit: Instead of directly discarding incorrect reasoning trajectories, this approach detects whether potential correct answers exist within them, thereby expanding the data instances..
2. Intrinsic Diversity for overthinking: The method analyzes the model's internal states to identify repetitive or redundant reasoning patterns, filtering such data to ensure conciseness.

Building on this, the authors integrate HSIR with GRPO and evaluate it on medical reasoning and arithmetic tasks, demonstrating the effectiveness of their approach.

**Strengths:**

1. This paper identifies key factors that influence performance improvement in model self-evaluation and proposes the HSIR method to enhance the quality of synthetic data generated through self-improvement. The authors subsequently integrate HSIR with GRPO for optimization.

2. Experiments conducted on the MedQA and GSM8K datasets demonstrate that the proposed approach outperforms several naive self-improvement baselines.

**Weaknesses:**

1. The contributions of this paper are limited, as the proposed methods rely heavily on existing approaches and lack core novelty.
    - The "verify-then-exit" strategy—which involves identifying and utilizing correct segments from failed reasoning trajectories—is a relatively common technique. Moreover, determining which parts of a failed trajectory are correct remains a challenging issue. If we solely rely on the appearance of the correct final answer as the criterion, this approach can only cover a narrow range of cases and is inadequate for most scenarios.
    - InDiv Score: Although labeled as "intrinsic," the idea of using hidden model states to quantify diversity is not novel. Simply incorporating attention weights does not constitute an insightful contribution. Furthermore, using attention weights as an intrinsic metric is a strong prior assumption, and its validity requires more thorough analysis and justification.
    - H-GRP: This is an incremental modification of GRPO that merely incorporates the InDiv score as an additional signal. The rationale behind this design and the reliability of the resulting reward signal remain questionable.

2. The experimental evaluation in the paper is insufficient. The main experiments only use MedQA and GSM8K—two relatively simple datasets. For out-of-distribution evaluation, the authors included Medbullets and MATH 500 datasets. Overall, the selection of datasets is limited and skewed toward lower difficulty. This lack of comprehensive experiments raises doubts about the method's effectiveness in realistically challenging reasoning tasks.

**Questions:**

please refer to weaknesses

---

> ### Author Response · Authors · 2025-11-23
> **Response to Reviewer DUEw (1/2)**
>
> We thank the reviewer for the insightful comments and are glad that he/she recognizes the motivation of our work and the effectiveness of our method. The responses to the questions are below:
>
> > W1. The "verify-then-exit" strategy—which involves identifying and utilizing correct segments from failed reasoning trajectories—is a relatively common technique. Moreover, determining which parts of a failed trajectory are correct remains a challenging issue. If we solely rely on the appearance of the correct final answer as the criterion, this approach can only cover a narrow range of cases and is inadequate for most scenarios.
>
> We emphasize that the goal of our VeriExit is to alleviate the data imbalance problem in the context of self-improvement training. To ensure data balance, existing methods usually allocate more trials to the difficult queries but often overlook the potential of reusing prior failed solutions. **In contrast, our VeriExit strategy innovatively salvages valid initial reasoning from failed attempts, providing a significant efficiency gain over costly resampling from scratch.** Thus, while VeriExit bears some resemblance to prior early-exit decoding methods, **its idea of reusing failed solutions for efficient data synthesis within self-improvement training is novel**.
>
> Moreover, to enhance the robustness of VeriExit, we also propose two more sophisticated variants: **NLI-based** and **prompt-based** strategies, which more accurately identify correct reasoning steps. Details and experimental results are provided in **Q2** of the **General Response**. In short, by leveraging an NLI model or prompting the LLM to evaluate the relationship between reasoning outputs and the final answer, these strategies can improve self-improvement performance. However, they would introduce considerable inference overhead. Therefore, for simplicity and efficiency, we adopt the text-matching method as the default VeriExit strategy. We have added this experiment and analysis to Appendix C1 of the revised paper.
>
> ---
>
> > W2. InDiv Score: Although labeled as "intrinsic," the idea of using hidden model states to quantify diversity is not novel. Simply incorporating attention weights does not constitute an insightful contribution. Furthermore, using attention weights as an intrinsic metric is a strong prior assumption, and its validity requires more thorough analysis and justification.
>
> Since the internal states of LLMs contain rich linguistic knowledge, it is relatively common to use them to probe the behavior of LLMs. However, **the core idea of our InDiv method, i.e., leveraging the intrinsic diversity among intermediate reasoning steps to quantify overthinking in reasoning LLMs, is new in the field of self-improvement training**. Furthermore, considering that some important tokens with higher attention weights may contribute more to the reasoning process, we introduce an attention-aware weighting mechanism to obtain more fine-grained intrinsic diversity. **The importance of tokens with higher attention weights has also been recognized by prior studies** [1-3].
>
> To validate the effectiveness of the attention-aware weighting mechanism, we compare our HSIR with a variant, “**-w/o attention-aware**”, which removes the attention-aware weighting mechanism by setting $Atten^k_{i,u}=\frac{1}{m}$ in Eq.5. The following table presents the comparative results of Qwen2.5-1.5B and Qwen2.5-3B models. We find that **removing the attention-aware weighting mechanism in InDiv leads to performance degradation**, demonstrating its contribution to InDiv. We have added this experiment and analysis to Appendix C2 of the revised paper.
>
> | Method | Qwen2.5-1.5B |  | Qwen2.5-3B |  |
> |---|:---:|:---:|:---:|:---:|
> |  | MedQA | GSM8K | MedQA | GSM8K |
> | SFT-Initial | 38.10 | 63.99 | 49.02 | 77.18 |
> | HSIR-SFT | 45.33 | 71.72 | 55.70 | 86.13 |
> | -w/o attenion-aware | 45.20 | 71.34 | 55.38 | 85.75 |

---

> > ### Author Response · Authors · 2025-11-23
> > **Response to Reviewer DUEw (2/2)**
> >
> > > W3. H-GRPO: This is an incremental modification of GRPO that merely incorporates the InDiv score as an additional signal. The rationale behind this design and the reliability of the resulting reward signal remain questionable.
> >
> > We would like to clarify that **the goal of H-GRPO is not to propose a new state-of-the-art GRPO method**, but rather to *examine whether our InDiv can be incorporated into GRPO to effectively mitigate the over-thinking problem* and address the performance degradation induced by existing length-oriented rewards. Therefore, we simply incorporate InDiv as an additional reward signal in the vanilla GRPO. In future work, we plan to explore more sophisticated integrations of VeriExit and InDiv into GRPO to achieve further improvements in both efficiency and performance. We have clarified it in Appendix A of our revised paper.
> >
> > ---
> >
> > > W4. The experimental evaluation in the paper is insufficient. The main experiments only use MedQA and GSM8K—two relatively simple datasets. For out-of-distribution evaluation, the authors included Medbullets and MATH 500 datasets.
> >
> > Following your suggestion, we further evaluate our HSIR on additional reasoning benchmarks: **ARC for scientific reasoning, CommonsenseQA for commonsense reasoning, and StrategyQA for multi-hop reasoning**. The experimental results are presented in **Q3** of the **General Response**. Overall, these results show that our HSIR still consistently outperforms the other counterparts among all benchmarks, demonstrating its generality.
> >
> > For out-of-distribution evaluation, we additionally include two challenging OOD test sets: **MedXpertQA** [4] for MedQA and **AMC2023** [5] for GSM8K. The following table reports the OOD performance of Qwen2.5-7B models trained with different self-improvement methods. As shown, **models enhanced with HSIR achieve superior OOD performance across all settings**, further validating the effectiveness of our approach. These additional OOD results have been included in Figure 4 of the revised paper.
> >
> > | MedXpertQA | STaR | ReST$^{EM}$ | HSIR-SFT | IRPO | HISR-DPO |
> > |---|---|---|---|---|---|
> > | Iteration 1 | 14.35 | 13.8 | 14.91 | 15.6 | 16.77 |
> > | Iteration 2 | 14.82 | 14.53 | 14.98 | 16.56 | 17.32 |
> > | Iteration 3 | 15.11 | 15.27 | 15.69 | 17.46 | 19.12 |
> >
> > | AMC2023 | STaR | ReST$^{EM}$ | HSIR-SFT | RPO | HISR-DPO |
> > |---|---|---|---|---|---|
> > | Iteration 1 | 40 | 42.5 | 45 | 47.5 | 50 |
> > | Iteration 2 | 42.5 | 47.5 | 52.5 | 50 | 52.5 |
> > | Iteration 3 | 45 | 52.5 | 55 | 55 | 57.5 |
> >
> > ---
> >
> > ### Reference
> >
> > [1] DRAGIN: Dynamic Retrieval Augmented Generation based on the Information Needs of Large Language Models, ACl2024. \
> > [2] DELTA: Dynamic Layer-Aware Token Attention for Efficient Long-Context Reasoning, arxiv2025. \
> > [3] Attention Illuminates LLM Reasoning: The Preplan-and-Anchor Rhythm Enables Fine-Grained Policy Optimization, arxiv2025. \
> > [4] MedXpertQA: Benchmarking Expert-Level Medical Reasoning and Understanding, ICML2025. \
> > [5] American Mathematics Competitions, 2023.

---

> ### Author Response · Authors · 2025-11-28
>
> Dear Reviewer DUEw,
>
> I hope this message finds you well. As the discussion period is nearing its end with **less than one week remaining**, we would like to ensure that we have addressed all your concerns satisfactorily. If you have any additional comments or feedback for us to consider, please feel free to let us know. Your insights are invaluable to us, and we are eager to resolve any remaining issues to further improve our work.
>
> Thank you very much for your time and effort in reviewing our paper.
>
> Best regards, \
> The authors of Paper 22671

---

### Official Review · Reviewer_nMXs · 2025-11-02

**Soundness:** 3
**Presentation:** 3
**Contribution:** 3
**Rating:** 6
**Confidence:** 3

**Summary:**

The paper proposes HSIR, a self-improvement recipe for reasoning LLMs that tackles two pain points of iterative self-training: data imbalance and overthinking. It introduces VeriExit, a verify-then-exit trajectory recycling strategy that truncates failed traces at the first correct step to harvest additional correct solutions on hard queries, and InDiv, an intrinsic diversity score computed from hidden states to filter verbose or redundant rationales; the pipeline is used for SFT and DPO, and extended to RL via H-GRPO that adds InDiv as a reward to encourage concise reasoning. Across benchmarks, the approach reports substantial gains in both accuracy and efficiency, with up to +10.9 percent average performance improvement and up to 42.4 percent reduction in inference overhead.

**Strengths:**

The paper is strong on originality by combining an intrinsic diversity criterion from hidden states with a verify-then-exit policy and a hierarchical GRPO variant, yielding a coherent recipe that reduces redundant reasoning without sacrificing accuracy. Methodologically the work is careful and high quality: it isolates contributions of InDiv and VeriExit with targeted ablations, tracks verifiable coverage to explain gains, and reports efficiency with detailed token accounting alongside accuracy. Clarity is high due to a clean problem framing, modular decomposition of the pipeline, and intuitive figures that map the training and inference loops to concrete decisions.

**Weaknesses:**

The paper’s core ideas are promising, but several issues limit clarity, generality, and fairness.

- The “verify-then-exit” procedure depends on ground-truth answers to scan intermediate steps and insert an exit token, which weakens the claim of fully self-improving training and risks overfitting to answer formats rather than robust reasoning; it also resembles prior verifier or early-exit style decoding and needs a tighter novelty delineation relative to answer-guided sampling and verifier-augmented methods.
- The Intrinsic Diversity score is computed from internal hidden states and attention weights, may be architecture and layer dependent, and lacks sensitivity analyses for layer choice, token selection, and thresholding; please report ablations on layer depth, normalization, and calibration across models, along with any failure modes when attention is diffuse.

- Efficiency is measured primarily by token count, not wall-clock latency or energy, so it is unclear whether HSIR reduces real inference cost; add throughput and latency on standardized hardware and include the overhead of InDiv and VeriExit.

- Comparisons could be fairer: VeriExit uses answer access while some baselines do not, iteration budgets and sampling counts differ across methods, and variance or significance is missing; please control total decoding tokens, report seeds and confidence intervals, and add stronger baselines such as verifier-trained STaR or RLVR variants with non-length rewards.

**Questions:**

How does VeriExit operate when gold answers are unavailable or noisy, and can you show robustness with imperfect verifiers or distant supervision?

What are the key sensitivity factors for InDiv (layer choice, token pooling, normalization, thresholds), and do results hold across architectures and model sizes?

Can you report wall-clock latency, throughput, and energy including the overhead of InDiv and VeriExit, not just token counts?

Are K and J chosen under a fixed decoding budget, and how do performance and stability vary with these hyperparameters across multiple seeds with confidence intervals?

---

> ### Author Response · Authors · 2025-11-23
> **Response to Reviewer nMXs (1/3)**
>
> We thank the reviewer for the positive and valuable comments and are glad that he/she recognizes the novelty of our method, and appreciates the comprehensiveness and high quality of our experiments. The responses to the questions are below:
>
> > W1. The “verify-then-exit” procedure depends on ground-truth answers to scan intermediate steps and insert an exit token, which weakens the claim of fully self-improving training and risks overfitting to answer formats rather than robust reasoning; it also resembles prior verifier or early-exit style decoding and needs a tighter novelty delineation relative to answer-guided sampling and verifier-augmented methods.
>
> First, we respectfully remind you that **in many prior studies [1-4] related to self-improvement training, ground-truth answers are also available**. The goal of self-improvement training is to encourage the model to self-generate the intermediate reasoning steps and leverage them to enhance the model’s reasoning capabilities. In this setting, the ground-truth is used to select the correct solutions from multiple candidate solutions. Even when the ground-truth answers are unavailable in some scenarios, we can follow [5] and **use the majority-voting answer of multiple self-generated solutions as the pseudo label for the VeriExit process**. The additional experiment in **Q1** of the **General Response** demonstrates that HSIR can still perform well in such unlabeled scenarios.
>
> Second, regarding the correction between VeriExit and prior early-exit style decoding methods, we acknowledge that VeriExit shares some conceptual resemblance. However, our approach introduces **a novel idea of reusing previous failed solutions for efficient data synthesis in the context of self-improvement training**. We have clarified it in the Related Work of the revised paper.
>
> ---
>
> > W2. The Intrinsic Diversity score is computed from internal hidden states and attention weights, may be architecture and layer dependent, and lacks sensitivity analyses for layer choice, token selection, and thresholding; please report ablations on layer depth, normalization, and calibration across models, along with any failure modes when attention is diffuse.
>
> Following your suggestion, we conduct additional ablation experiments to **investigate the impact of key components in our InDiv**. Specifically, for layer choice, we use the Qwen2.5-1.5B model (with a total of 28 layers) as an example and vary the layer depth for calculating InDiv scores across {5, 10, 15, 20, 25}. The comparative results on MedQA are presented in the following table. For reference, we also report the results of SFT-Initial and ReST$^{EM}$. Notably, all models are self-improved for one iteration.
>
> | Method | MedQA |
> |-|-|
> | SFT-Initial | 38.1 |
> | ReST$^{EM}$ | 41.63 |
> | HSIR-SFT |  |
> | 5th layer | 43.43 |
> | 10th layer | 43.91 |
> | 15th layer | 45.33 |
> | 20th layer | 45.19 |
> | 25th layer | 44.23 |
>
> As seen, **HSIR-SFT consistently outperforms the other baseline methods across different layer choices**, indicating that its effectiveness is not highly sensitive to layer depth. Moreover, using the middle layer (i.e., 15th layer) yields the best performance. We conjecture that the middle layer encodes richer and more useful semantic information, thus resulting in more accurate InDiv scores. Therefore, we choose to use the middle layer for calculating InDiv scores in this work.
>
> For the attention-aware weighting mechanism and InDiv normalization, we evaluate their impact by comparing two HSIR variants: 1) “**-w/o attention-aware**”, which removes the attention-aware weighting mechanism by setting $Atten^k_{i,u}=\frac{1}{m}$ in Eq.5; 2) “**-w/o normalization**”, which directly uses the $\mathbf{InDiv}^k_i$ from Eq.5 as the final score and filters the solutions with lower scores for each query. Notably, we adjust the filtering ratio in both variants to ensure that the training data size is approximately the same as in the original HSIR-SFT method. The following table presents the comparative results, from which we find that removing either component of InDiv leads to performance degradation. These results consistently prove the effectiveness of both strategies. We have added these experiments to Appendix C2 of the revised paper.
>
> | Method | Qwen2.5-1.5B |  | Qwen2.5-3B |  |
> |-|:-:|:-:|:-:|:-:|
> |  | MedQA | GSM8K | MedQA | GSM8K |
> | SFT-Initial | 38.10 | 63.99 | 49.02 | 77.18 |
> | HSIR-SFT | 45.33 | 71.72 | 55.70 | 86.13 |
> | -w/o attenion-aware | 45.20 | 71.34 | 55.38 | 85.75 |
> | -w/o normalization | 43.52 | 71.27 | 54.13 | 85.60 |
>
> Regarding the threshold $\tau$ of InDiv, we respectfully remind you that **we had analyzed its effect in Appendix E4 of the original paper**, which has been updated in Appendix C3 of the revised paper. As for the tokens used to compute InDiv scores, we emphasize that all tokens from the self-generated solution are included without any selection. We have clarified it in Section 3.2 of our revised paper.

---

> > ### Author Response · Authors · 2025-11-23
> > **Response to Reviewer nMXs (2/3)**
> >
> > > W3. Efficiency is measured primarily by token count, not wall-clock latency or energy, so it is unclear whether HSIR reduces real inference cost; add throughput and latency on standardized hardware and include the overhead of InDiv and VeriExit.
> >
> > Since all models are evaluated on the same hardware, **the number of generated tokens can reflect the wall-clock inference latency in our experiments**. Regarding the overhead of VeriExit, we only sample the final answer without regenerating the intermediate reasoning trajectories, which is much faster than simply allocating more trials for failed queries. In our preliminary experiments, we found that the VeriExit sampling can be finished in **approximately one hour on 8 NVIDIA A800 (80GB) GPUs**, which is about 1/4 of the time required to generate full reasoning trajectories.
> >
> > For InDiv, the query and its solution are fed into the model in a teacher-forcing manner, requiring only a single forward pass and minimal latency. In practice, during HSIR-SFT and HSIR-DPO training, obtaining hidden states for each token introduces some computational overhead, but this can be completed in **about half an hour on 8 NVIDIA A800 (80GB) GPUs**. Moreover, during H-GRPO training, we can reuse the hidden representation obtained by the reference model without extra forward passes. **The actual computation of the InDiv score itself is lightweight, involving only simple vector operations that take a few seconds**. We have clarified these details in Appendix C7 of our revised paper.
> >
> > ---
> >
> > > W4. Comparisons could be fairer: VeriExit uses answer access while some baselines do not, iteration budgets and sampling counts differ across methods, and variance or significance is missing; please control total decoding tokens, report seeds and confidence intervals, and add stronger baselines such as verifier-trained STaR or RLVR variants with non-length rewards.
> >
> > We should emphasize that for **all baseline methods in our experiments also have access to the ground-truth answers**, which are used to select the correct candidate solutions for self-improvement training. Thus, we believe that the comparisons are fair. **Regarding the iteration budgets, we follow the original settings of each baseline**. Specifically, since RFT and ReGenesis are non-iterative methods, we perform only one iteration of self-improvement training. Notably, for all baselines, we maintain a fixed data synthesis budget. For instance, for the non-iterative RFT method, we follow [6] and increase the sampling counts (from K to $T \times K$) to ensure consistency in the sampling budget. We have clarified it in Section 4.1 of our revised paper.
> >
> > Regarding the stronger baselines, we respectively remind you that the STaR method used in our experiment **already relies on answer verification**, i.e., using the ground-truth answer to select correct solutions. While for the RLVR variants with non-length rewards, we emphasize that **our goal is not to propose a new state-of-the-art GRPO method**, but to examine whether our proposed InDiv method can be incorporated into GRPO to effectively mitigate the over-thinking problem and address the performance degradation induced by existing length-oriented rewards. Therefore, we only compare our H-GRPO with two representative GRPO algorithms that rely on length-oriented rewards. We have clarified it in Appendix A of our revised paper.

---

> > > ### Author Response · Authors · 2025-11-23
> > > **Response to Reviewer nMXs (3/3)**
> > >
> > > > Q1. How does VeriExit operate when gold answers are unavailable or noisy, and can you show robustness with imperfect verifiers or distant supervision?
> > >
> > > This question is closely related to W1. Please **refer to our response to W1** for details.
> > >
> > > ---
> > >
> > > > Q2. What are the key sensitivity factors for InDiv (layer choice, token pooling, normalization, thresholds), and do results hold across architectures and model sizes?
> > >
> > > This question is closely related to W2. Please **refer to our response to W2** for details.
> > >
> > > ---
> > >
> > > > Q3. Can you report wall-clock latency, throughput, and energy including the overhead of InDiv and VeriExit, not just token counts?
> > >
> > > This question is closely related to W3. Please **refer to our response to W3** for details.
> > >
> > > ---
> > >
> > > > Q4. Are K and J chosen under a fixed decoding budget, and how do performance and stability vary with these hyperparameters across multiple seeds with confidence intervals?
> > >
> > > Yes, the hyperparameters K and J are set to 10 and 5, respectively, and remain fixed across all experiments. Here, the K denotes the sampling count used by all self-improvement methods, which is not introduced by our method. Therefore, we only investigate the impact of J in this experiment. Specifically, **we evaluate our HSIR-SFT with different J values ranging from 1 to 7**, while keeping K=10. The comparative results of Qwen2.5-1.5B on MedQA are presented in the following table. For reference, we also include the results of SFT-Initial and ReST$^{EM}$. All models are self-improved for one iteration.
> > >
> > > | Method | MedQA |
> > > |---|---|
> > > | SFT-Initial | 38.1 |
> > > | ReST$^{EM}$ | 41.63 |
> > > | HSIR-SFT |  |
> > > | J=1 | 42.11 |
> > > | J=2 | 42.66 |
> > > | J=3 | 43.36 |
> > > | J=4 | 45.16 |
> > > | J=5 | 45.33 |
> > > | J=6 | 43.75 |
> > > | J=7 | 43.28 |
> > >
> > > From these results, we find that: 1) When J is too small (e.g., J=1), our VeriExit struggles to sample enough correct solutions, limiting the effectiveness of HSIR. 2) When J is too large (e.g., J=7), there are too many similar VeriExit-sampled solutions that share the same prefix reasoning steps. These similar solutions would reduce the diversity of training data and lead to sub-optimal results. 3) **Across all J values, our HSIR consistently outperforms the baseline methods, proving its robustness**. Notably, in the case of J=5, HSIR achieves the best performance, thus leaving it as the default setting in our work. We have added this experiment to Appendix C3 of the revised paper.
> > >
> > > ---
> > >
> > > ### Reference
> > >
> > > [1] Star: Bootstrapping reasoning with reasoning, NeurIPS2022. \
> > > [2] Scaling Relationship on Learning Mathematical Reasoning with Large Language Models, arxiv2023. \
> > > [3] Self-Training with Direct Preference Optimization Improves Chain-of-Thought Reasoning, ACL2024. \
> > > [4] Iterative reasoning preference optimization, NeurIPS2024. \
> > > [5] Large Language Models Can Self-Improve, EMNLP2023. \
> > > [6] V-STaR: Training Verifiers for Self-Taught Reasoners, COLM2024.

---

> ### Author Response · Authors · 2025-11-28
>
> Dear Reviewer nMXs,
>
> I hope this message finds you well. As the discussion period is nearing its end with **less than one week remaining**, we would like to ensure that we have addressed all your concerns satisfactorily. If you have any additional comments or feedback for us to consider, please feel free to let us know. Your insights are invaluable to us, and we are eager to resolve any remaining issues to further improve our work.
>
> Thank you very much for your time and effort in reviewing our paper.
>
> Best regards, \
> The authors of Paper 22671

---

### Official Review · Reviewer_kaYb · 2025-11-06

**Soundness:** 3
**Presentation:** 3
**Contribution:** 3
**Rating:** 6
**Confidence:** 3

**Summary:**

The paper proposes HSIR, a self-improvement framework for Large Reasoning Models that targets two observed issues in self-training: data imbalance and overthinking. HSIR adds (i) VeriExit, a “verify-then-exit” decoding that scans previously failed traces and, once an intermediate step reaches the gold answer, truncates and resamples the final answer; and (ii) an Intrinsic Diversity (InDiv) score computed from model hidden states to filter repetitive traces. On MedQA and GSM8K with multiple LLMs, the authors report up to +10.9% average accuracy gains and –42.4% relative inference “overhead” . They also extend InDiv as an auxiliary reward in H-GRPO.

**Strengths:**

+ The paper crisply diagnoses imbalance/overthinking, and proposes two pragmatic fixes (trajectory recycling + intrinsic diversity) that are easy to add to SFT/DPO pipelines.

+ HSIR improves Qwen2.5 1.5B/3B/7B under SFT and DPO; it also outperforms STaR/ReSTEM and reduces tokens per output; broader one-iteration SFT results on Qwen3-1.7B, Phi-3.5-mini, Mistral-7B, and LLaMA3-8B support generality.

+ Useful ablations for VeriExit vs. answer-driven sampling and InDiv vs. length penalties; extension to H-GRPO shows accuracy preserved while keeping responses shorter relative to GRPO baselines.

**Weaknesses:**

- The pipeline relies on gold final answers to verify intermediate steps in VeriExit; robustness of the “step reaches the gold” heuristic (string match like “answer is {y}”) is unclear.

- Seed data are distilled from DeepSeek-R1 (and an alternative QWQ-32B), which may bias comparisons and complicate conclusions about self-improvement vs. teacher quality. Stronger decontamination/controls would help.

- Tasks are only MedQA and GSM8K; OOD checks are small (Medbullets/MATH). Broader reasoning (coding, multi-hop QA, planning) would strengthen generality.

- Baselines are not representative: key inference-time (SC/best-of-N/early-stop), verifier-rerank, and “SC→finetune” curation baselines are missing.

**Questions:**

as in weakness

---

> ### Author Response · Authors · 2025-11-23
> **Response to Reviewer kaYb (1/2)**
>
> We thank the reviewer for the positive and insightful comments and are glad that he/she recognizes the values of our method and appreciates the comprehensiveness of our experiments. The responses to the questions are below:
>
> > W1. The pipeline relies on gold final answers to verify intermediate steps in VeriExit; robustness of the “step reaches the gold” heuristic (string match like “answer is {y}”) is unclear.
>
> First, we emphasize that, following many prior studies[1][2][3][4], we assume the ground-truth answer $y_i$ is available in our work. Even when the ground-truth answers are unavailable, we can use the majority-voting answer among multiple self-generated solutions as the pseudo answer to perform the VeriExit process. The additional experiment presented in **Q1** of the **General Response** demonstrates that HSIR can still perform well in unlabeled scenarios. We have added this experiment to Appendix C5 of the revised paper.
>
> Second, regarding the robustness of  VeriExit, we provide a detailed response in **Q2** of the **General Response**. In short, we show that the simple text-matching strategy exhibits strong alignment with more sophisticated NLI-based and prompt-based alternatives. Although the latter two methods can result in better performance, they will lead to more inference overhead. Thus, for simplicity and efficiency, we adopt the text-matching method in our work by default. We have added this experiment and analysis to Appendix C1 of the revised paper.
>
> ---
>
> > W2. Seed data are distilled from DeepSeek-R1 (and an alternative QWQ-32B), which may bias comparisons and complicate conclusions about self-improvement vs. teacher quality. Stronger decontamination/controls would help.
>
> We would like to clarify that **all baseline methods use the same seed data**, ensuring fair and unbiased comparisons in our experiments.  Nevertheless, following your suggestions, we conduct additional experiments to analyze the impact of different sources of seed data. Specifically, we additionally focus on two types of seed data: 1) **low-quality seed data**, which is distilled from a smaller and weaker LRMs; 2) **self-distilled seed data**, which is generated by the model itself via an in-context learning strategy.
>
> Using the Qwen2.5-3B as the testbed, we adopt the SFT-Initial Qwen2.5-1.5B to produce the low-quality seed data. For the self-distilled seed data, we randomly select three DeepSeek-distilled examples as few-shot demonstrations and use them to guide the base Qwen2.5-3B to generate the seed data. Notably, for all methods, we use the same set of queries and sample a correct solution for each query. That is, we keep the number of seed training samples consistent. The Qwen2.5-3B model is first fine-tuned on each type of seed data, and then self-improved using various SFT approaches for one iteration. The comparative results on MedQA are reported in the following table.
>
> | Method | DeepSeek-R1 Seed | QwQ-32B Seed | Qwen2.5-1.5B Seed | Self-distill Seed |
> |:---:|:---:|:---:|:---:|:---:|
> | SFT-Initial | 49.02 | 50.04 | 42.50 | 48.70 |
> | STaR | 49.25 | 49.25 | 43.36 | 47.53 |
> | ReST$^{EM}$ | 55.22 | 54.77 | 47.18 | 49.80 |
> | HSIR-SFT | 55.70 | 55.70 | 48.00 | 50.17 |
>
> From these results, we observe that: 1) **the quality of seed data is indeed important**, as initial SFT with low-quality seed data will lead to substantial performance degradation; 2) **among all types of seed data, our HSIR can consistently bring performance gains**, continuing to prove its effectiveness. Overall, while it is feasible for an LLM to self-distill seed data and subsequently self-improve using HSIR, leveraging more high-quality seed data from stronger frontier models allows HSIR to realize its full potential and achieve better performance. We have added this experiment and analysis to Appendix C4 of our revised paper.

---

> > ### Author Response · Authors · 2025-11-23
> > **Response to Reviewer kaYb (2/2)**
> >
> > > W3. Tasks are only MedQA and GSM8K; OOD checks are small (Medbullets/MATH). Broader reasoning (coding, multi-hop QA, planning) would strengthen generality.
> >
> > Following your suggestion, we evaluate our HSIR on additional reasoning benchmarks: ARC for scientific reasoning, CommonsenseQA for commonsense reasoning, and StrategyQA for multi-hop reasoning. The experimental results are presented in **Q3** of the **General Response**. In summary, our HSIR consistently outperforms the other counterparts across these benchmarks, demonstrating its broad generality.
> >
> > For OOD evaluation, we further include two challenging OOD test sets: **MedXpertQA** [5] for MedQA and **AMC2023** [6] for GSM8K. The following table reports the OOD results of Qwen2.5-7B models trained with different self-improvement methods. As shown, with the help of our HSIR methods, the trained models can achieve better OOD results across all settings, highlighting the effectiveness of HSIR. These additional OOD results have been included in Figure 4 of the revised paper.
> >
> > | MedXpertQA | STaR | ReST$^{EM}$ | HSIR-SFT | IRPO | HISR-DPO |
> > |---|---|---|---|---|---|
> > | Iteration 1 | 14.35 | 13.8 | 14.91 | 15.6 | 16.77 |
> > | Iteration 2 | 14.82 | 14.53 | 14.98 | 16.56 | 17.32 |
> > | Iteration 3 | 15.11 | 15.27 | 15.69 | 17.46 | 19.12 |
> >
> > | AMC2023 | STaR | ReST$^{EM}$ | HSIR-SFT | RPO | HISR-DPO |
> > |---|---|---|---|---|---|
> > | Iteration 1 | 40 | 42.5 | 45 | 47.5 | 50 |
> > | Iteration 2 | 42.5 | 47.5 | 52.5 | 50 | 52.5 |
> > | Iteration 3 | 45 | 52.5 | 55 | 55 | 57.5 |
> >
> > ---
> >
> > > W4. Baselines are not representative: key inference-time (SC/best-of-N/early-stop), verifier-rerank, and “SC→finetune” curation baselines are missing.
> >
> > As our primary goal is to propose a self-improvement training method, we do not compare HSIR with inference-time strategies in the main experiments. Nevertheless, following your suggestion, we **additionally compare our HSIR with the inference-time self-consistency (SC) method**. Specifically, during inference, we sample 5 solutions from the model and select the majority-voting answer as the final prediction. The following table shows the comparative results of Qwen2.5 models on MedQA, where ``-w/ SC'' means using the SC method.
> >
> > | Method | Qwen2.5-1.5B | Qwen2.5-3B | Qwen2.5-7B |
> > |---|---|---|---|
> > | SFT-Initial | 38.10 | 49.02 | 62.45 |
> > | SFT-Initial -w/SC | 45.64 | 56.94 | 67.44 |
> > | HSIR-SFT | 46.58 | 57.58 | 68.74 |
> > | HSIR-SFT -w/SC | 49.49 | 61.27 | 72.19 |
> >
> > As seen, by increasing the test-time compute, the SC method indeed enhances the reasoning performance of SFT-Initial models effectively. However, **it still underperforms our HSIR-SFT method**, even if HSIR-SFT only samples a single solution during inference. More encouragingly, **combining the HSIR-SFT and SC methods consistently yields further performance improvements**. For instance, for the Qwen2.5-3B model, with the help of SC, HSIR-SFT achieves a 12.25% performance gain over the SFT-Initial model. These results prove the compatibility of HSIR with the inference-time SC method. We have added this experiment to Appendix C9 of our revised paper.
> >
> > Regarding the “SC→finetune” baseline, we implement it by using the majority-voting answer among multiple candidate solutions as the pseudo label and perform the subsequent self-improvement training. Please refer to **Q1** of the **General Response** for more detailed experimental results and analyses. It is noteworthy that since the verifier-rerank baseline requires an additional verifier model and increased inference budget during model evaluation, which falls outside the scope of this work, we do not include it in our comparisons.
> >
> > ---
> >
> > ### Reference
> >
> > [1] Star: Bootstrapping reasoning with reasoning, NeurIPS2022. \
> > [2] Scaling Relationship on Learning Mathematical Reasoning with Large Language Models, arxiv2023. \
> > [3] Self-Training with Direct Preference Optimization Improves Chain-of-Thought Reasoning, ACL2024. \
> > [4] Iterative reasoning preference optimization, NeurIPS2024. \
> > [5] MedXpertQA: Benchmarking Expert-Level Medical Reasoning and Understanding, ICML2025. \
> > [6] American Mathematics Competitions, 2023.

---

> ### Author Response · Authors · 2025-11-28
>
> Dear Reviewer kaYb,
>
> I hope this message finds you well. As the discussion period is nearing its end with **less than one week remaining**, we would like to ensure that we have addressed all your concerns satisfactorily. If you have any additional comments or feedback for us to consider, please feel free to let us know. Your insights are invaluable to us, and we are eager to resolve any remaining issues to further improve our work.
>
> Thank you very much for your time and effort in reviewing our paper.
>
> Best regards, \
> The authors of Paper 22671

---

### Author Response · Authors · 2025-11-23
**General Response (1/3)**

We thank all reviewers for their valuable comments and suggestions. All comments have been carefully considered and we have made corrections based on them. **The submission has been updated, and the corrections are marked in blue in the revised paper.** Specifically, regarding the general comments of reviewers, we provide the following responses and analyses.

### **Q1. Does HSIR work when ground-truth answers are unavailable?**

Following many prior studies [1-4], we assume that the ground-truth answers are available in this work. We acknowledge that in real-world applications, the ground-truth answers might be unavailable. In such cases, we can follow [5] and **use the majority-voting answer among multiple candidate solutions as a pseudo label**. Although these pseudo labels may contain errors, our HSIR still has the potential to improve the LRMs. To examine its effectiveness under this setting, we conduct experiments on several Qwen2.5 models and present the results in the following table.

| Methods | Qwen2.5-1.5B |  | Qwen2.5-3B |  | Qwen2.5-7B |  | Overall |  |
|---|:---:|:---:|:---:|:---:|:---:|:---:|:---:|:---:|
|  | MedQA | GSM8K | MedQA | GSM8K | MedQA | GSM8K | Acc | Tokens |
| SFT-Initial | 38.10 | 63.99 | 49.02 | 77.18 | 62.45 | 83.93 | 62.45 | 1,536 |
| ReST$^{EM}$ | 41.63 | 69.75 | 55.22 | 83.95 | 64.18 | 88.17 | 67.15 | 1,268 |
| HSIR | 45.33 | 71.72 | 55.70 | 86.13 | 67.32 | 88.78 | 69.16 | 1,075 |
| SC->ReST$^{EM}$ | 40.46 | 70.36 | 51.69 | 83.70 | 64.57 | 87.04 | 66.30 | 1,322 |
| SC->HSIR-SFT | 44.78 | 71.72 | 54.99 | 84.69 | 65.91 | 89.46 | 68.59 | 1,108 |

For reference, we use the ReST$^{EM}$ as the compared method, and also report the results based on ground-truth answers. Notably, “SC->” refers to using the pseudo labels for self-improvement training. As shown, training with pseudo labels indeed yields sub-optimal performance compared with using ground-truth answers. Nevertheless, our HSIR-SFT method still brings significant performance improvements against the SFT-Initial model, i.e., +6.14% average performance gains, proving that **HSIR can still perform well in the unlabeled scenarios**. We have added this experiment to Appendix C5 of the revised paper.

---

> ### Author Response · Authors · 2025-11-23
> **General Response (2/3)**
>
> ### **Q2. The robustness of VeriExit**
>
> The core of our VeriExit is to determine whether the reasoning step arrives at the ground-truth answer. There are multiple ways to achieve this. Specifically, the simplest approach is to directly match the text between the reasoning output and the ground-truth answer. We acknowledge that such a text-matching method could not be robust enough, as the reasoning step may imply the final answer without explicitly matching its text. To address this limitation, inspired by SelfCheckGPT[6], **we further propose two more sophisticated VeriExit strategies: an NLI-based approach and a prompt-based approach**. In the NLI-based strategy, we use an external Natural Language Inference (NLI) model to judge the relationships between the reasoning output and the answer. In the prompt-based strategy, we query the current model $\mathcal{M}_{t-1}$ using a carefully-designed prompt to assess whether the reasoning output and answer convey the same meaning.
>
> To investigate the effectiveness of these strategies, we calculate the coverage of verifiable successful solutions on the most difficult queries of MedQA using different VeriExit strategies. Using the text-matching method as the baseline, we measure its correlation with NLI-based and prompt-based methods via the Pearson Correlation Coefficient (PCC). In this experiment, we use the Qwen2.5-3B model. As shown, **the text-matching method exhibits relatively strong correlations with both alternative methods**, with PCC scores exceeding 0.4 and p-values below 0.05. These results suggest that all VeriExit variants produce broadly consistent predictions.
>
> | Setting | PCC | p-value |
> |---|:---:|:---:|
> | Text-match -> NLI-based | 0.4 | 8.60E-39 |
> | Text-match -> Prompt-based | 0.51 | 5.40E-68 |
>
> Moreover, we replace the VeriExit strategy in the HSIR-SFT framework, and compare the performance and efficiency of HSIR-SFT variants in the following table. All experiments are conducted using the Qwen2.5-3B model. The results reveal that: 1) **both NLI-based and prompt-based methods can achieve better reasoning performance**, as they can more accurately identify the correct reasoning steps during VeriExit; 2) **despite their effectiveness, these two methods lead to more inference overhead**. Therefore, for simplicity and efficiency, we adopt the text-matching method in our work by default. We have added this experiment and analysis to Appendix C1 of the revised paper.
>
> | Method | MedQA | Inference time per query |
> |---|---|---|
> | Text-matching | 55.7 | 0.007s |
> | NLI-based | 56.48 | 1.449s |
> | Prompt-based | 58.05 | 3.506s |

---

> ### Author Response · Authors · 2025-11-23
> **General Response (3/3)**
>
> ### **Q3. Evaluation on more reasoning tasks**
>
> To further validate the generality of our HSIR, we evaluate our HSIR on additional reasoning tasks. Specifically, we use the **AI2 Reasoning Challenge's challenge set (ARC)**[7] for scientific reasoning, **CommonsenseQA (CSQA)**[8] for commonsense reasoning, and **StrategyQA (StraQA)**[9] for multi-hop reasoning. DeepSeek-R1 is used to collect the seed data for each task. The comparative results of Qwen2.5 models are reported in the following table. Notably, we conduct one iteration of self-improvement SFT training in this experiment.
>
> | Method |  | Qwen2.5-1.5B |  |  | Qwen2.5-3B |  |  | Qwen2.5-7B |  | Overall |  |
> |:---:|:---:|:---:|:---:|:---:|:---:|:---:|:---:|:---:|:---:|:---:|:---:|
> |  | ARC | StrategyQA | CSQA | ARC | StrategyQA | CSQA | ARC | StrategyQA | CSQA | Accuracy | Tokens |
> | SFT-Initial | 68.33 | 59.10 | 60.03 | 79.91 | 66.67 | 71.91 | 86.43 | 72.05 | 73.63 | 70.90 | 1,298 |
> | SFT-Oracle | 69.38 | 60.84 | 74.20 | 80.66 | 69.29 | 79.69 | 87.55 | 74.38 | 82.47 | 75.38 | 1,223 |
> | STaR | 68.58 | 61.28 | 62.74 | 81.20 | 68.85 | 73.05 | 87.98 | 71.76 | 74.86 | 72.26 | 1,232 |
> | ReST$^{EM}$   | 69.78 | 62.30 | 68.14 | 83.26 | 68.59 | 77.64 | 89.18 | 72.36 | 79.19 | 74.49 | 1,148 |
> | HSIR-SFT | **72.45** | **64.63** | **72.48** | **83.92** | **69.59** | **78.54** | **89.87** | **73.65** | **81.65** | **76.31** | **996** |
>
>
> As seen, **our HSIR-SFT consistently outperforms the other counterparts across all benchmarks by a clear margin**. More specifically, compared to the base model, HSIR-SFT brings +5.41% average performance gains and reduces 23.3% average inference tokens. These results can prove the generality of our method. We have added this experiment to Section 4.3 of the revised paper.
>
> ---
>
> ### Reference
>
> [1] Star: Bootstrapping reasoning with reasoning, NeurIPS2022. \
> [2] Scaling Relationship on Learning Mathematical Reasoning with Large Language Models, arxiv2023. \
> [3] Self-Training with Direct Preference Optimization Improves Chain-of-Thought Reasoning, ACL2024. \
> [4] Iterative reasoning preference optimization, NeurIPS2024. \
> [5] Large Language Models Can Self-Improve, EMNLP2023. \
> [6] Selfcheckgpt: Zero-resource black-box hallucination detection for generative large language models, EMNLP2023. \
> [7] Think you have solved question answering? try arc, the ai2 reasoning challenge, arxiv2018. \
> [8] Commonsenseqa: A question answering challenge targeting commonsense knowledge, NAACL2019. \
> [9] Did aristotle use a laptop? a question answering benchmark with implicit reasoning strategies, TACL2021.

---

### Author Response · Authors · 2025-11-30
**Summary**

We thank all the reviewers for their constructive comments and suggestions.  We are glad that reviewers found the paper **well-written and easy to follow** (tSEv, nMXs), recognized the promise of our approach in **improving performance across various post-training paradigms** (kaYb, DUEw, tSEv), and acknowledged **the originality and overall quality of the work** (nMXs). We address some common feedback here as well as provide an overview of what we have updated in the manuscript.

- **More reasoning benchmarks and OOD evaluation** (kaYb, DUEw, tSEv): We evaluate our HSIR on additional reasoning benchmarks: ARC [1] for scientific reasoning, CommonsenseQA [2] for commonsense reasoning, and StrategyQA [3] for multi-hop reasoning. For the OOD evaluation in Figure 4, we additionally include two challenging OOD test sets: MedXpertQA [4] for MedQA and AMC2023 [5] for GSM8K.  In summary, these additional results show that our HSIR consistently outperforms the other counterparts across all benchmarks, demonstrating its broad generality. We have added the results of more reasoning benchmarks to Section 4.3 and updated the OOD results in Figure 4 of the revised manuscript.
- **Analysis of the robustness of VeriExit** (kaYb, nMXs, DUEw): We improve the robustness of VeriExit by introducing two more sophisticated strategies (NLI-based and prompt-based) to robustly determine whether the reasoning step arrives at the ground-truth answer. The additional experiments and analyses have been shown in Appendix C1 of the revised manuscript.
- **HSIR without using ground-truth answers** (kaYb, nMXs): In this setting, we follow the prior study [6] and use the majority-voting answer among multiple candidate solutions as a pseudo label to perform our HSIR. The additional results show that HSIR still outperforms the others and brings significant performance gains, i.e., +6.14% average performance gains. We have added this experiment to Appendix C5 of the revised paper.
- **Analysis of the seed data** (kaYb, tSEv): We further investigate the influence of seed data by additionally using two types of seed data: low-quality seed data and self-distilled seed data. We have added this experiment and analysis to Appendix C4 of the revised manuscript.
- **More ablation and parameter analyses** (nMXs, tSEv): We conduct more ablation experiments to examine the effect of important components in InDiv, and investigate the influence of two important parameters: 1) layer depth for computing the InDiv score, and 2) sampling count J for collecting correct solutions. The additional results and analyses have been shown in Appendix C2 and C3 of the revised manuscript.
- **Analysis of the efficiency of HSIR** (nMXs, tSEv): We have added more details about the wall-clock time of important components of our HSIR in Appendix C7 of the revised manuscript.
- **Difference between HSIR and prior studies** (nMXs, DUEw, tSEv): We have explained the difference between our HSIR and prior works, and clarified the contributions of our work in the Related Work section of the revised manuscript.

---

### Reference:
[1] Think you have solved question answering? try arc, the ai2 reasoning challenge, arxiv2018. \
[2] Commonsenseqa: A question answering challenge targeting commonsense knowledge, NAACL2019. \
[3] Did aristotle use a laptop? a question answering benchmark with implicit reasoning strategies, TACL2021. \
[4] MedXpertQA: Benchmarking Expert-Level Medical Reasoning and Understanding, ICML2025. \
[5] American Mathematics Competitions, 2023. \
[6] Large Language Models Can Self-Improve, EMNLP2023.

---

### Meta-Review · Area_Chair_AQbu · 2025-12-30

**Summary:**

The core concerns include insufficient novelty of HSIR’s key components (VeriExit, InDiv Score, H-GRPO), which rely heavily on existing approaches; inadequate experimental evaluation with limited, low-difficulty datasets (MedQA, GSM8K) and insufficient OOD tests; and unsubstantiated design rationales (e.g., InDiv’s dependency on attention weights). These issues undermine the work’s contribution and generalizability.

**Reviewer Concerns:**

Addressed concerns: Insufficient experiments (additional reasoning benchmarks/ OOD tests added); VeriExit’s robustness (NLI/prompt-based variants validated); HSIR’s effectiveness without ground-truth (pseudo-label experiments).

Outstanding concerns: Core novelty clarification (VeriExit/InDiv/H-GRPO still incremental)

**Reviewer Scores:**

Reviewer DUEw (original score: 2/reject) would likely not raise the score to accept. The rebuttal addressed experimental gaps but failed to resolve the core novelty and design justification issues, which were the reviewer’s primary concerns about the work’s contribution.

---

### Decision · Program_Chairs · 2026-01-26

Reject